# DiL: Discrete-anchored Representation Alignment for Semi-Supervised Continual Learning

**Nanyi Wang** [1]  **Chaojie Chen** [1]  **Zuoqi Tang** [1]  **Jinxiang Lai** [2]  **Xingcai Wu** [1]  **Qi Wang** [1]

## Abstract

Leveraging the unlabeled stream is crucial yet challenging in Semi-Supervised Continual Learning (SSCL) under continual class expansion. Existing SSCL methods typically enforce dense pseudo-label consistency and indiscriminate distillation on unlabeled data, which can reinforce errors and intensify base–novel interference. To address these issues, we propose Discrete-anchored Incremental Learning (DiL) to ground continual updates on reliable discrete anchors that remain stable under noisy pseudo-labels. DiL introduces Discrete Contrastive Distillation (DCD), which discretizes the distillation pathway and performs anchor-referenced selective distillation to curb error reinforcement. Meanwhile, Class-Aware Channel-Chunked Encoding (CACE) learns channel-chunked representations and exploits the confusion matrix induced by the discrete anchors to separate novel from confusable base classes. Extensive experiments on multiple datasets show that DiL achieves state-of-the-art performance across diverse SSCL protocols.

## 1. Introduction

Continual learning (CL) has been introduced to incrementally acquire new knowledge while maintaining previously learned knowledge across tasks (De Lange et al., 2021; Shi et al., 2025). In realistic settings, labels are often scarce, whereas unlabeled data are abundant, which motivates semi-supervised continual learning (SSCL) to extend CL by exploiting unlabeled data to learn robust representations. To address forgetting in SSCL, many methods

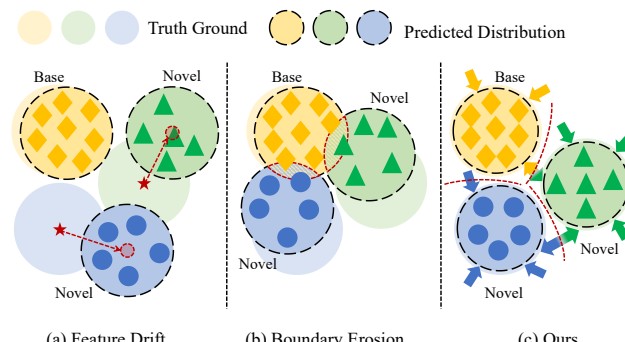

*Figure 1.* Comparison with existing SSCL paradigms. (a) Feature drift: Full-dimensional alignment enforces compactness but propagates pseudo-label noise, shifting predictions from the ground truth. (b) Boundary erosion: Unsupervised expansion can ignore base-novel dynamics, causing disordered growth that blurs class boundaries. (c) Our method: Reduces drift from noisy pseudo labels and increases separation between classes.

adopt replay-based or generative strategies for incremental learning (Wang et al., 2021), and others use pseudo-label-driven consistency to strengthen supervision on unlabeled data (Smith et al., 2021). Recently, USP (Duan et al., 2025) anchors unlabeled distillation to stable class means to improve memory stability under continual class expansion. These works further improve both accuracy and stability under scarce supervision across long task sequences.

Since unlabeled data constitute most of the SSCL training set, their effective use largely determines performance, and prior work broadly falls into two main lines. One line pairs pseudo-labeling with distillation or consistency constraints to exploit unlabeled data while alleviating forgetting (Smith et al., 2021; Fan et al., 2024). Nevertheless, they often assume that all dimensions are equally informative, thus enforcing consistency across all feature channels, which can amplify pseudo-label noise into **feature drift** (Fig. 1 (a)). The other line leverages unlabeled data to regularize class representations, encouraging compact prototypes and separable geometry (Liu et al., 2024; Duan et al., 2025). Yet such regularization is rarely aware of class confusions, fails to separate novel classes from confusable base classes, thereby inducing **boundary erosion** (Fig. 1 (b)). Overall, existing approaches typically emphasize either feature stability or

[1]Guizhou Provincial Laboratory of Big Data, College of Computer Science and Technology, Guizhou University, Guiyang, China [2]The Hong Kong University of Science and Technology, Hong Kong, China. Correspondence to: Qi Wang <qiwang@gzu.edu.cn>.

*Proceedings of the 43rd International Conference on Machine Learning*, Seoul, South Korea. PMLR 306, 2026. Copyright 2026 by the author(s).

geometric separation, while both issues may co-occur under class-incremental settings and lead to **feature drift** or **boundary erosion**.

To reduce **feature drift** induced by label noise, prior work beyond SSCL employs indirect denoising in noisy label learning. Representative methods include RankMatch (Zhang et al., 2023) and OT-Filter (Feng et al., 2023), which select reliable samples to build robust supervision, guided by confidence or distribution geometry. However, selecting reliable samples does not guarantee correctness, which can cause erroneous signals to be repeatedly reinforced during subsequent training. To address **boundary erosion** under class expansion, some methods impose explicit separation in representation space, e.g., ECOC with discrete codewords (Dietterich & Bakiri, 1995; Chou & Chen, 2025) or CREATE with subspace partitioning (Chen et al., 2025). The separation approaches often rely on a static separation scheme and fail to adapt to emerging class confusions. In light of these observations, we argue that an effective SSCL framework should suppress error reinforcement while preserving class separation under continual expansion (Fig. 1 (c)). In this paper, we distill only informative channels to curb noise accumulation, then employ dynamic confusion constraints to reduce class confusion, consequently mitigating feature drift and boundary erosion.

To this end, we propose Discrete-anchored Incremental Learning (DiL), a unified framework that builds class-conditioned anchors in a discrete space through two components: Discrete Contrastive Distillation (DCD) and Class-Aware Channel-Chunked Encoding (CACE). Specifically, DCD takes the discrete anchors as references to identify channels aligned with class prototypes, distilling only relevant corresponding dimensions. Meanwhile, CACE computes similarity between anchors to derive a confusion matrix and assigns chunk combinations for novel classes guided by the prior. With this assignment, the selected chunk combinations align novel instances with their anchors, which enlarges the separation from confusable existing classes. Experiments across diverse SSCL settings show consistent gains for DiL, including a 9.16% boost in last task accuracy on the 20-task ImageNet-100 benchmark with only 1% labels. Our contributions are summarized as follows:

- We propose a unified SSCL framework that grounds class-incremental learning on discrete anchors to suppress noise propagation and reduce class confusion.

- We design an effective distillation mechanism in DCD, which applies mutual information guided channel selection to suppress noisy gradients from pseudo labels.

- We introduce an allocation method in CACE that places novel classes farther from confusable base classes for proactive isolation.

**Conflict of Interest Disclosure**  The authors declare no financial conflicts of interest.

## 2. Related Works

### 2.1. Continual Learning

Continual learning (CL) studies how to learn from a data stream while preserving previously learned knowledge, and a wide range of approaches have been developed to mitigate catastrophic forgetting (Zhou et al., 2024). Replay-based methods rehearse stored samples to stabilize training, as in experience replay (ER) (Rolnick et al., 2019). With episodic memory, gradient projection methods further constrain parameter updates to reduce interference, as in GEM and its efficient variant AGEM (Lopez-Paz & Ranzato, 2017; Chaudhry et al., 2019). Replay can also be generative, which synthesizes past data with learned models, including DGR (Shin et al., 2017) and diffusion-based replay (Gao & Liu, 2023). Distillation methods transfer knowledge from old to new models through consistency constraints, such as LwF (Li & Hoiem, 2017). Dynamic network methods mitigate interference by expanding or isolating model capacity, as exemplified by PNN (Rusu et al., 2016) and Pack-Net (Mallya & Lazebnik, 2018). Although effective, dense alignment across all channels in distillation-based CL may propagate unstable dimensions, so our method selectively constrains key channels for robustness.

### 2.2. Semi-supervised Learning

Semi-supervised learning (SSL) exploits unlabeled data under scarce annotations to improve generalization (Yang et al., 2022). Early work, such as the $\Pi$-model (Laine & Aila, 2017), promotes consistency, and subsequent methods mainly follow consistency regularization and pseudo-label-based self-training. Mean Teacher (Tarvainen & Valpola, 2018) and VAT (Miyato et al., 2018) improve consistency learning via teacher–student targets and adversarial perturbations, while Pseudo-Label (Lee et al., 2013) extends supervision with high-confidence predictions. Recent frameworks such as MixMatch (Berthelot et al., 2019), UDA (Xie et al., 2020), and FixMatch (Sohn et al., 2020) further enhance unlabeled utilization with augmentation-driven consistency and thresholded pseudo-labels. Robust variants, including FlexMatch (Zhang et al., 2022) and FreeMatch (Wang et al., 2022), improve reliability under imbalanced utilization or noisy conditions. When SSL is applied to more realistic class-incremental settings, distribution shifts can intensify error reinforcement over time.

### 2.3. Semi-supervised Continual Learning

Semi-supervised continual learning (SSCL) learns from a task stream with few labels, abundant unlabeled data, and

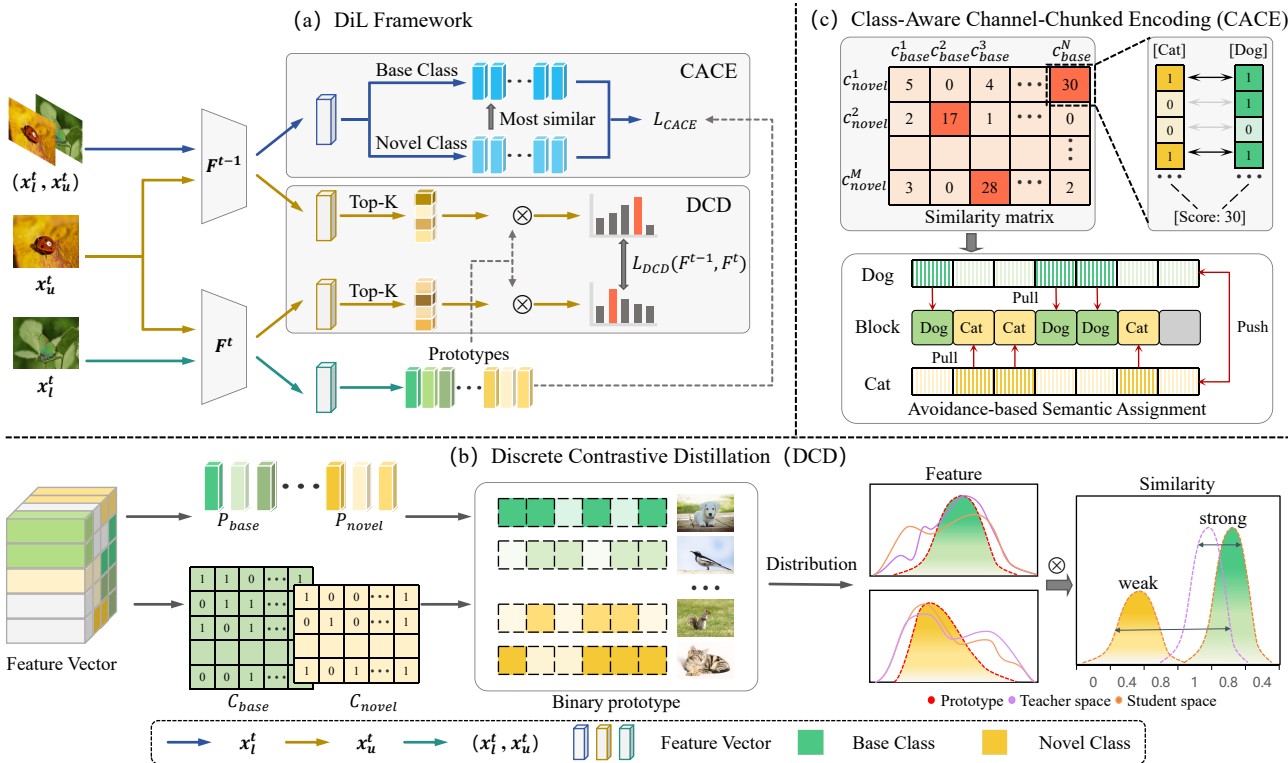

Figure 2. Overview of the DiL framework. At task $t$, the current stream contains labeled samples $x_l^t$ and unlabeled samples $x_u^t$, and training follows a teacher-student paradigm with the previous model frozen. The detailed training workflow is provided in Section 3.2.

an expanding class set. DistillMatch (Smith et al., 2021) combines consistency regularization with knowledge distillation to exploit unlabeled data while alleviating forgetting. ORDisCo (Wang et al., 2021) stabilizes prior knowledge through replay-style synthesis and prediction consistency. Brahma et al. (Brahma et al., 2021) introduce hypernetworks for parameter generation and transfer during task evolution. Recent methods mainly strengthen the robustness of unlabeled supervision. CCIC (Boschini et al., 2022b) and NNCSL (Kang et al., 2023) enhance cross-view constraints or representation learning. UaD-CIE (Cui et al., 2023) incorporates uncertainty-aware learning in few-shot settings. DSGD (Fan et al., 2024) improves robustness with dynamic subgraph distillation, while USP (Duan et al., 2025) adopts a divide-and-conquer framework for more reliable unlabeled learning and cross-task stability. Different from these SSCL methods, our method achieves stronger robustness under scarce labels with a unified design that couples reliable distillation and confusion-aware assignment.

## 3. Method

### 3.1. Preliminaries

Under SSCL, we consider a sequence of tasks indexed by $t$, where at each task $t$, a labeled set $D_l^t = \{(x_l, y_l)\}$ and

an unlabeled set $D_u^t = \{x_u\}$ are provided. The model contains a feature extractor $f_\theta : \mathcal{X} \to \mathbb{R}^d$ and a classifier $g_\theta$. The class set observed up to task $t$ is $y_{\text{all}}^t = \{1, \ldots, C_{\text{all}}^t\}$, where $y_{\text{b}}^t = \{1, \ldots, C_{\text{base}}^t\}$ denotes the base classes and $y_{\text{n}}^t = \{C_{\text{base}}^t + 1, \ldots, C_{\text{all}}^t\}$ denotes the novel classes. For each class $c$, we maintain a prototype $P_c \in \mathbb{R}^d$ as the mean teacher feature $z_{\theta_{t-1}}(x)$ over $x \in D_{l,c}^t$, later used to construct discrete anchor prototypes. To mitigate forgetting, a replay buffer $E^t$ stores samples from previous tasks. A teacher-student paradigm is adopted, where the teacher is the frozen previous model $\theta_{t-1}$ and the student is $\theta_t$.

**Supervised loss.** For labeled samples, the supervised loss is defined as cross-entropy on weak augmentation $a_w(\cdot)$:

$$\mathcal{L}_{\text{sup}}(D_l^t) = \mathbb{E}_{(x_l^t, y_l^t) \sim D_l^t} H\big(y_l^t, \, p_{\theta_t}(a_w(x_l^t))\big), \quad (1)$$

where $H(\cdot, \cdot)$ denotes the standard cross-entropy loss and $p_{\theta_t}(x) = \text{softmax}(g_{\theta_t}(f_{\theta_t}(x)))$.

**Unsupervised loss.** Following FixMatch (Sohn et al., 2020), pseudo-labels are obtained from weak predictions. Then, the unsupervised loss is defined as:

$$\mathcal{L}_{\text{uns}}(D_u^t) = \mathbb{E}_{x_u^t \sim D_u^t}\Big[\mathbb{I}(\hat{p}_u^t \geq \tau) \, H\big(\hat{y}_u^t, \, p_s^t\big)\Big], \quad (2)$$

where $p_w^t = p_{\theta_t}(a_w(x_u^t))$ and $p_s^t = p_{\theta_t}(a_s(x_u^t))$ are the predictions under weak and strong augmentations. The

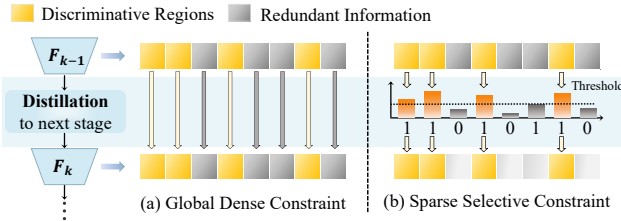

(a) Global Dense Constraint | (b) Sparse Selective Constraint

*Figure 3.* Illustration of the feature constraint mechanism. Baselines enforce dense full-dimensional alignment, whereas DCD performs sparse distillation on selected dimensions for robustness.

pseudo-label and its confidence are $\hat{y}_u^t = \arg\max p_w^t$ and $\hat{p}_u^t = \max p_w^t$. $\tau$ is the confidence threshold in $\mathbb{I}(\hat{p}_u^t \geq \tau)$.

**Replay distillation.** To preserve previously learned knowledge, replay distillation is applied to $E^t$, with replay examples $x_e^t$ uniformly sampled from the buffer during training. Then, the distillation loss is defined as:

$$\mathcal{L}_{\text{cl}}(E^t) = \tau_{kd}^2 \, \mathbb{E}_{x_e^t \sim E^t} \text{KL}\Big(p_{\theta_{t-1}}^{\tau_{kd}}(x_e^t) \,\big\|\, p_{\theta_t}^{\tau_{kd}}(x_e^t)\Big), \quad (3)$$

where $\ell_\theta(x) = g_\theta(f_\theta(x))$ denotes the logits. The predictive distribution is defined as $p_\theta^{\tau_{kd}}(x) = \text{softmax}(\ell_\theta(x)/\tau_{kd})$ with temperature $\tau_{kd}$ (Hinton et al., 2015). After task $t$, the student $\theta_t$ is frozen and used as the teacher for task $t+1$.

### 3.2. Overview

As shown in Fig. 2 (a), task $t$ follows a teacher-student scheme, where the previous model $F^{t-1}$ is frozen and the student $F^t$ is updated. Ground-truth labels optimize $\mathcal{L}_{\text{sup}}$ in Eq. (1), with pseudo-labels supervising unlabeled samples through the consistency loss $\mathcal{L}_{\text{uns}}$ in Eq. (2). In parallel, $D_l^t$ is processed by DCD to estimate class-wise importance masks and to construct discrete anchor prototypes in Eq. (5) for channel-selected distillation. Distillation over the reliable set in Eq. (6) then defines $\mathcal{L}_{\text{DCD}}$. In this work, we define a confusion matrix $M_{\text{conf}} = C_{\text{novel}} C_{\text{base}}^{\top}$ by binarizing the similarity scores $\rho(n, c) = \langle \hat{P}_n, \hat{P}_c \rangle$ computed from masked prototypes (Eq. (4)). The confusion matrix further guides block assignment in CACE to compute $\mathcal{L}_{\text{CACE}}$ in Eq. (14). All losses are aggregated in Eq. (15) to update $F^t$.

### 3.3. Discrete Contrastive Distillation

Distillation is widely used in continual learning to alleviate forgetting (Li & Hoiem, 2017; Rebuffi et al., 2017; Douillard et al., 2020; Hinton et al., 2015), and it typically enforces dense teacher-student alignment by matching logits over all classes or aligning the full feature representation across dimensions (Fig. 3 (a)). However, under SSCL, pseudo-label noise can propagate through distillation and further exacerbate representation drift. To address this problem, we propose Discrete Contrastive Distillation (DCD) (Fig. 3 (b)), which performs channel-selected distillation guided by

class-conditioned discrete anchor prototypes. DCD consists of an Importance Mask to build masked prototype targets, and Partitioned Distillation imposes class-aware distillation.

**Importance Mask.** For each class $c$, class-level importance is estimated from labeled samples $D_{l,c}^t$ only, which avoids bias from noisy pseudo-labels. Given teacher features $z_{\theta_{t-1}}(x) \in \mathbb{R}^d$, a class-specific importance vector $M_c \in \mathbb{R}^d$ is computed. Each entry $M_{c,j}$ scores the $j$-th channel for class $c$ by combining (i) average activation magnitude and (ii) activation frequency. After normalization, applied after aggregation to ensure comparable channel scales across classes, the importance score is defined as:

$$M_c = w_I \mathcal{N}\Big(\mathbb{E}_{x \sim D_{l,c}^t}\big[\,|z_{\theta_{t-1}}(x)|\,\big]\Big) \\ + w_F \mathcal{N}\Big(\mathbb{E}_{x \sim D_{l,c}^t}\big[\mathbb{I}(|z_{\theta_{t-1}}(x)| > \beta)\big]\Big), \quad (4)$$

where $\mathcal{N}(\cdot)$ denotes $\ell_2$ normalization, with $w_I = w_F = \frac{1}{2}$ to balance magnitude and frequency terms. The threshold $\beta$ is class-specific and is set to the mean absolute activation over labeled samples of class $c$. We then define $m_c \in \{0, 1\}^d$ by selecting $\Omega_c = \text{TopK}(M_c, K_{\text{top}})$ and setting $m_{c,j} = \mathbb{I}[j \in \Omega_c]$. In practice, the impact of $K_{\text{top}}$ is studied in Fig. 8 (b) across datasets of different scales. With this mask, the discrete anchors prototype and its stacked form are defined as:

$$\hat{P}_c = \frac{P_c \odot m_c}{\|P_c \odot m_c\|_2 + \epsilon}, \\ \hat{P} = [\hat{P}_1, \ldots, \hat{P}_{C_{\text{all}}^t}]^{\top} \in \mathbb{R}^{C_{\text{all}}^t \times d}, \quad (5)$$

where $\epsilon$ is a small constant added for numerical stability to avoid degenerate normalization. These prototypes serve as discrete anchors to define the confusion matrix and to construct masked distillation targets in subsequent modules.

**Partitioned Distillation.** With the discrete prototype $\hat{P}$ from class-level masks, we perform discrete class partitioned alignment in key channel subspaces. For distillation, each class $c$ is associated with a reliable sample set that combines labeled samples and high-confidence unlabeled samples. The reliable sample set is defined as:

$$D_c = D_{l,c}^t \cup D_{u,c}^t, \\ D_{u,c}^t = \{x_u \in D_u^t \mid \hat{y}_u^t(x_u) = c, \ \hat{p}_u^t(x_u) \geq \tau\}. \quad (6)$$

In addition to the class-level selection, we derive a sample-level mask that emphasizes the most active channels for the current input. For each input sample $x$, we define a sample level active channel set $\Omega_x$ via $\text{TopK}(|z_\theta(x)|, K_{\text{top}})$, adopting the same $K_{\text{top}}$ as in the class-level selection to keep a consistent sparsity budget, which yields a binary mask $m_{x,j} = \mathbb{I}[j \in \Omega_x]$. To softly suppress unselected dimensions while preserving gradient flow, a leaky masking operator is applied:

$$\delta(z; \alpha) = z \odot m_x + \alpha\big(z \odot (1 - m_x)\big), \quad (7)$$

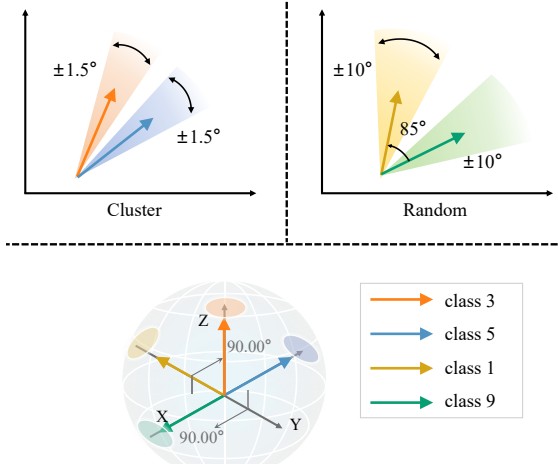

*Figure 4.* Cosine similarity measuring. We employ a unit hypersphere model to analyze confusing CIFAR-10 pairs, Cat–Dog (3,5) and Auto–Truck (1,9), where cosine similarity is measured as the angle between $\ell_2$-normalized class prototypes.

where $\alpha \in (0, 1)$ is a decay coefficient that reduces the contribution of unselected dimensions while keeping their gradients nonzero. Cosine similarity logits are computed from the masked feature $\delta(z_\theta(x); \alpha)$ against the discrete prototypes $\hat{P}$:

$$s_\theta(x) = \frac{\delta(z_\theta(x); \alpha)\,\hat{P}^\top}{\|\delta(z_\theta(x); \alpha)\|_2 + \epsilon}. \tag{8}$$

The distillation distribution is obtained as $q_\theta(x) = \text{softmax}(s_\theta(x))$. To explicitly balance stability and plasticity under class expansion, the distillation objective is partitioned over base and novel classes. For a class subset $y \subseteq y_{\text{all}}^t$, the renormalized sub-distribution is defined as $\tilde{q}_\theta(x)[c] = \frac{q_\theta(x)[c]}{\sum_{c' \in y} q_\theta(x)[c']}$ for $c \in y$. The DCD objective is defined as:

$$\begin{aligned}
\mathcal{L}_{\text{DCD}}(x) = {}& w_{\text{b}}\,\text{KL}\Big(\tilde{q}_{\theta_{t-1}}(x)[y_{\text{b}}^t] \,\Big\|\, \tilde{q}_{\theta_t}(x)[y_{\text{b}}^t]\Big) \\
& + w_{\text{n}}\,\text{KL}\Big(\tilde{q}_{\theta_{t-1}}(x)[y_{\text{n}}^t] \,\Big\|\, \tilde{q}_{\theta_t}(x)[y_{\text{n}}^t]\Big).
\end{aligned} \tag{9}$$

where $w_{\text{b}}$ enforces strict preservation of base-class alignment and $w_{\text{n}}$ allows adaptive learning for novel classes, with their specific choices reported in Appendix C.4.2.

### 3.4. Class-Aware Channel-Chunked Encoding

To maintain separability under continual class expansion, recent methods (Liu et al., 2024; Duan et al., 2025; Chen et al., 2025) introduce explicit regularization to stabilize representations. Nevertheless, these approaches lack mechanisms for dynamically identifying confusable classes. Inspired by prototype anchoring (Li et al., 2024; Ji et al., 2025) and

the ECOC (Dietterich & Bakiri, 1995; Chou & Chen, 2025) perspective of representing classes with discrete codewords, CACE constructs block-sparse chunk assignments to form near-orthogonal class anchors in the discrete space (Fig. 4). Specifically, CACE partitions features into disjoint chunks to enlarge margins and then leverages a confusion matrix to select block combinations.

**Block Partition Space Construction.** First, we construct non-overlapping block partitions and block-sparse compositional representations to limit inter-class block-support overlap and enlarge effective decision margins. In particular, the $d$-dimensional feature space is partitioned into $B$ consecutive, non-overlapping chunks, each containing $\frac{d}{B}$ dimensions. The index set of the $b$-th block is denoted as $\mathcal{B}_b \subset \{1, \ldots, d\}$, satisfying:

$$\begin{aligned}
& \mathcal{B}_b \cap \mathcal{B}_{b'} = \varnothing \ (b \neq b'), \\
& \bigcup_{b=1}^{B} \mathcal{B}_b = \{1, \ldots, d\}.
\end{aligned} \tag{10}$$

Each class is assigned a block-sparse chunk support set $S_c \subseteq \{1, \ldots, B\}$ with $|S_c| = k$, which induces the coordinate subspace indexed by $\bigcup_{b \in S_c} \mathcal{B}_b$. Due to disjoint chunks, the activated subspace spans $d_{\text{eff}} = k\frac{d}{B}$ dimensions, and all valid $k$-block patterns constitute:

$$\mathcal{E} = \{S \subseteq \{1, \ldots, B\} \mid |S| = k\}. \tag{11}$$

The detailed selection and derivation of $B$ and $k$ are deferred to Appendix A.

**Semantics-Driven Avoidance Assignment.** Instead of random allocation, novel class block assignments are determined by avoiding confusable base classes, guided by the masked prototypes in Eq. (5). For a novel class $n$ and a base class $c \in y_{\text{b}}^t$, the confusion score is defined as $\rho(n, c) = \langle \hat{P}_n, \hat{P}_c \rangle$ based on masked prototypes. For each $n \in y_{\text{n}}^t$, the resulting confusion set is defined as:

$$\mathcal{N}_{\text{conf}}(n) = \arg \max_{\substack{\mathcal{S} \subseteq y_{\text{b}}^t \\ |\mathcal{S}| = K_{\text{conf}}}} \sum_{c \in \mathcal{S}} \rho(n, c). \tag{12}$$

We set $K_{\text{conf}} = 1$ by default, as focusing on the dominant confusion yields a strong avoidance cue while avoiding overly restrictive assignments. In the combinatorial space $\mathcal{E}$, the base-class chunk supports $\{S_c\}_{c \in y_{\text{b}}^t}$ are fixed. A novel-class chunk selection $S_n \in \mathcal{E}$ is obtained by minimizing its overlap with the confusion set:

$$S_n = \arg \min_{S \in \mathcal{E}} \sum_{c \in \mathcal{N}_{\text{conf}}(n)} |S \cap S_c|. \tag{13}$$

With $S_n$ determined, we construct class anchors by sampling Gaussian noise on the active chunks, masking the others

*Table 1.* Comparison of average and final accuracy on CIFAR-10 and CIFAR-100 (-X denotes labeled samples per class). Bold numbers indicate the best performance, and results for our method are averaged over 5 independent runs with different random seeds.

| Method | CIFAR10-30 | | CIFAR10-150 | | CIFAR100-20 | | CIFAR100-25 | | CIFAR100-80 | | CIFAR100-125 | |
|---|---|---|---|---|---|---|---|---|---|---|---|---|
| | Avg | Last | Avg | Last | Avg | Last | Avg | Last | Avg | Last | Avg | Last |
| iCaRL (Rebuffi et al., 2017) | 34.16 | 21.84 | 60.86 | 53.65 | 26.43 | 13.92 | 28.14 | 15.29 | 36.32 | 19.10 | 44.14 | 30.73 |
| DER (Yan et al., 2021) | 40.41 | 31.48 | 64.77 | 61.06 | 31.01 | 23.53 | 32.82 | 26.53 | 53.32 | 41.55 | 57.21 | 48.86 |
| CCIC (Boschini et al., 2022a) | – | 55.20 | – | 74.30 | – | – | – | 29.50 | – | – | – | 44.30 |
| ORDisCo (Wang et al., 2021) | – | – | 74.77 | 65.91 | – | – | – | – | – | – | – | – |
| NNCSL (Kang et al., 2023) | – | – | – | – | 55.19 | 43.53 | 57.45 | 46.00 | 67.27 | 55.35 | 67.58 | 56.40 |
| iCaRL&Fix (Fan et al., 2024) | 45.98 | 30.71 | 78.36 | 69.08 | 45.75 | 23.40 | 49.83 | 31.25 | 53.46 | 32.21 | 56.87 | 41.38 |
| +DSGD (Fan et al., 2024) | 77.33 | **76.41** | 84.14 | 79.69 | 52.80 | 35.47 | 53.42 | 35.95 | 57.92 | 37.81 | 58.08 | 43.14 |
| +USP (Duan et al., 2025) | 79.66 | 70.43 | 84.78 | 78.21 | 53.20 | 41.30 | 54.36 | 38.25 | 58.59 | 44.20 | 59.96 | 43.80 |
| **+DiL (Ours)** | 79.82 | 71.55 | 84.17 | 78.57 | 56.16 | 45.55 | 58.15 | 44.58 | 66.37 | 52.32 | 67.57 | 55.23 |
| DER&Fix (Fan et al., 2024) | 66.71 | 61.41 | 81.10 | 77.00 | 51.76 | 40.86 | 52.03 | 44.47 | 64.03 | 50.25 | 66.69 | 53.57 |
| +DSGD (Fan et al., 2024) | 75.04 | 72.59 | 83.08 | 79.39 | 55.63 | 44.63 | 57.94 | 46.68 | 65.48 | 55.40 | 69.14 | 58.50 |
| +USP (Duan et al., 2025) | **81.43** | 73.65 | 84.43 | 77.74 | 58.79 | 45.22 | 59.87 | 47.44 | 68.67 | 60.45 | 71.60 | 63.08 |
| **+DiL (Ours)** | 81.39 | 73.32 | **85.80** | **80.91** | **59.19** | **50.76** | **62.11** | **52.48** | **70.19** | **61.40** | **72.41** | **63.34** |

*Table 2.* Comparison of task-wise accuracy for base and novel classes on the 11-task CUB-200 benchmark, where Task 1 contains 100 fully supervised base classes, and each subsequent task introduces 10 novel classes with 5 labeled samples per class.

| Method | Classes | Task ID | | | | | | | | | | | Avg |
|---|---|---|---|---|---|---|---|---|---|---|---|---|---|
| | | 1 | 2 | 3 | 4 | 5 | 6 | 7 | 8 | 9 | 10 | 11 | |
| SS-iCaRL (Rebuffi et al., 2017) | Base | 69.89 | 62.32 | 60.62 | 58.99 | 58.59 | 57.77 | 59.88 | 56.21 | 54.46 | 50.54 | 46.11 | 57.76 |
| | Novel | – | 53.22 | 32.38 | 24.07 | 22.76 | 23.34 | 17.58 | 16.40 | 16.39 | 16.13 | 16.32 | 23.86 |
| SS-NCM-CNN (Mensink et al., 2012) | Base | 69.89 | 65.80 | 64.97 | 63.79 | 63.81 | 61.08 | 65.24 | 63.73 | 58.77 | 55.74 | 51.88 | 62.24 |
| | Novel | – | 56.37 | 34.70 | 26.03 | 24.04 | 24.68 | 19.14 | 18.60 | 17.70 | 17.79 | 18.36 | 25.74 |
| UaD-CIE (Cui et al., 2023) | Base | 75.87 | 74.58 | 74.09 | 73.46 | 72.24 | **71.68** | **71.33** | **70.50** | 70.15 | 69.27 | **69.13** | 72.03 |
| | Novel | – | 57.35 | 46.29 | 39.58 | 45.02 | 42.54 | 45.37 | 42.75 | 40.82 | 41.98 | 42.49 | 44.42 |
| + USP (Duan et al., 2025) | Base | 78.21 | 75.00 | 74.13 | 73.36 | **72.42** | 70.74 | 68.99 | 68.44 | 67.25 | 66.62 | 66.66 | 71.07 |
| | Novel | – | 69.18 | 60.07 | 50.93 | 55.67 | 53.08 | 52.86 | 53.64 | 51.07 | 53.00 | 54.57 | 55.41 |
| + DiL (Ours) | Base | **78.77** | **75.77** | **74.53** | **74.23** | 72.14 | 70.43 | 68.37 | 67.26 | 68.34 | 67.83 | 67.77 | 71.40 |
| | Novel | – | **75.50** | **73.39** | **72.26** | **70.57** | **68.48** | **67.13** | **64.13** | **65.43** | **63.40** | **61.87** | **68.22** |

to zero, and $\ell_2$-normalizing the result. The anchors are stacked as $A \in \mathbb{R}^{C_{\text{all}}^t \times d}$ and used as class-specific targets to supervise features from the reliable sets $\{D_c\}$ in Eq. (6). The CACE loss is defined on the feature representation $z = f_\theta(x)$ as:

$$\mathcal{L}_{\text{CACE}} = \mathbb{E}_{(x,c) \sim \cup_c D_c}\left[\ell_{\text{ce}}\left(\frac{z(x)}{\|z(x)\|_2}\left(\frac{A}{\|A\|_{\text{row},2}}\right)^\top, c\right)\right]. \tag{14}$$

where $c$ denotes the class index associated with sample $x$ in $D_c$. The overall objective is defined as:

$$\begin{aligned}\mathcal{L}_{\text{total}} = \ &\mathcal{L}_{\text{sup}} + \lambda_{\text{uns}}\mathcal{L}_{\text{uns}} + \lambda_{\text{cl}}\mathcal{L}_{\text{cl}} \\ &+ \lambda_{\text{dcd}}\mathcal{L}_{\text{DCD}} + \lambda_{\text{cace}}\mathcal{L}_{\text{CACE}},\end{aligned} \tag{15}$$

## 4. EXPERIMENTS

### 4.1. Experimental Settings

**Dataset.** We evaluate on CIFAR-10 (Krizhevsky et al., 2009), CIFAR-100 (Krizhevsky et al., 2009), ImageNet-100 (Deng et al., 2009), and CUB-200 (Chaudhry et al., 2018). CIFAR-10/100 contain 32×32 images with 50k training and 10k test samples, covering 10 and 100 classes, re-

spectively. ImageNet-100 is a 100-class subset of ImageNet-1K at 256×256 resolution, with about 1.3k training and 500 validation images per class on average. CUB-200 includes 200 fine-grained bird categories and is used under the few-shot SSCL setting with scarce labels.

**Task Settings.** We follow the class-incremental SSCL protocol used in DSGD (Fan et al., 2024) and USP (Duan et al., 2025) for fair comparison. For CIFAR-10/100 and ImageNet-100, classes are split into sequential tasks with 2, 10, and 10 novel classes per task, respectively. We evaluate multiple label budgets on each dataset and denote each setting as *Dataset*-X (e.g., CIFAR10-30). For CUB, we follow the UaD-CIE (Cui et al., 2023) few-shot SSCL protocol and, as in USP, implement our method in the official UaD-CIE codebase under the same training and evaluation pipeline.

**Metrics.** Following previous research on continual learning, we adopt the average incremental accuracy $A_{\text{Avg}}$ and the last-task accuracy $A_{\text{Last}}$ as the main evaluation metrics. For a sequence of $T$ tasks, $a_{t,i}$ denotes the top-1 test accuracy on task $i$ after finishing task $t$. After task $t$, the incremental accuracy is $A_t = \frac{1}{t}\sum_{i=1}^{t} a_{t,i}$, and $A_{\text{Avg}} = \frac{1}{T}\sum_{t=1}^{T} A_t$. In addition, we propose center drift to quantify the displacement magnitude of class centers between adjacent tasks

*Table 3.* Comprehensive task-wise accuracy breakdown on the 20-task ImageNet-100 continual learning benchmark, illustrating performance evolution across tasks under different labeled data ratios.

| Labels | Method | 1 | 2 | 3 | 4 | 5 | 6 | 7 | 8 | 9 | 10 | 11 | 12 | 13 | 14 | 15 | 16 | 17 | 18 | 19 | 20 | Avg |
|---|---|---|---|---|---|---|---|---|---|---|---|---|---|---|---|---|---|---|---|---|---|---|
| | NNCSL (Kang et al., 2023) | 59.50 | 50.20 | 39.71 | 43.50 | 38.58 | 34.13 | 32.88 | 29.50 | 30.59 | 29.84 | 27.93 | 30.53 | 31.09 | 30.37 | 30.22 | 29.70 | 29.62 | 29.36 | 28.79 | 28.98 | 34.25 |
| | iCaRL&Fix+USP(Duan et al., 2025) | 64.80 | 50.80 | 52.93 | 49.50 | 44.80 | 39.67 | 34.97 | 34.55 | 32.49 | 31.48 | 29.27 | 33.13 | 33.48 | 34.57 | 34.05 | 33.12 | 32.87 | 31.29 | 30.40 | 28.64 | 37.84 |
| 1% | DER&Fix+USP(Duan et al., 2025) | 64.40 | 55.00 | 53.33 | 51.10 | 47.12 | 43.13 | 41.60 | 41.00 | 38.58 | 37.08 | 35.64 | 38.10 | 37.78 | 36.91 | 36.53 | 34.20 | 33.48 | 33.09 | 33.64 | 32.78 | 41.22 |
| | iCaRL&Fix+**DiL** (Ours) | 68.20 | 56.75 | 55.47 | 52.98 | 51.84 | 45.62 | 41.57 | 44.05 | 41.64 | 37.60 | 38.32 | 35.97 | 31.90 | 34.32 | 33.40 | 34.98 | 31.19 | 31.31 | 30.68 | 28.64 | 41.50 |
| | DER&Fix+**DiL** (Ours) | **70.30** | **68.45** | **67.73** | **66.75** | **65.90** | **64.18** | **61.21** | **60.00** | **58.62** | **56.77** | **54.29** | **53.82** | **51.54** | **50.18** | **48.65** | **46.76** | **46.47** | **44.65** | **43.30** | **41.94** | **56.07** |
| | NNCSL (Kang et al., 2023) | 58.00 | 55.60 | 45.43 | 48.80 | 27.93 | 39.53 | 39.53 | 39.53 | 37.59 | 40.04 | 39.52 | 42.13 | 42.31 | 43.51 | 43.16 | 41.73 | 39.40 | 41.69 | 42.43 | 43.26 | 42.56 |
| | iCaRL&Fix+USP(Duan et al., 2025) | 73.60 | 62.40 | 68.00 | 66.00 | 61.52 | 56.93 | 54.80 | 52.55 | 51.11 | 51.84 | 50.04 | 52.23 | 51.85 | 52.11 | 52.40 | 50.85 | 49.81 | 49.16 | 49.05 | 48.46 | 54.56 |
| 5% | DER&Fix+USP(Duan et al., 2025) | 76.00 | 74.80 | 72.00 | 72.00 | 63.68 | 60.20 | 58.63 | 57.10 | 54.93 | 53.12 | 53.20 | 55.20 | 55.17 | 55.63 | 55.89 | 54.70 | 53.58 | 53.53 | 53.37 | 53.62 | 59.32 |
| | iCaRL&Fix+**DiL** (Ours) | 84.90 | **88.60** | 84.13 | 76.40 | 65.72 | 72.93 | 74.66 | 74.53 | **73.93** | 71.69 | 70.13 | 67.68 | 66.34 | 64.59 | 62.84 | 61.09 | 59.34 | 57.59 | 55.84 | 55.84 | 70.28 |
| | DER&Fix+**DiL** (Ours) | **86.20** | 87.65 | **86.00** | **85.03** | **82.70** | **76.56** | **76.66** | **75.53** | 69.84 | **73.86** | **71.54** | **68.81** | **69.84** | **68.56** | **67.59** | **65.80** | **64.01** | **61.22** | **60.43** | **58.64** | **72.82** |
| | NNCSL (Kang et al., 2023) | 60.00 | 60.00 | 51.43 | 54.30 | 48.17 | 43.40 | 42.12 | 41.90 | 44.05 | 44.44 | 42.33 | 44.03 | 45.53 | 46.14 | 45.78 | 46.24 | 43.53 | 41.48 | 41.67 | 44.12 | 46.53 |
| | iCaRL&Fix+USP(Duan et al., 2025) | 78.00 | 77.00 | 79.73 | 78.50 | 71.60 | 68.00 | 65.09 | 63.00 | 60.13 | 58.12 | 57.83 | 58.97 | 59.82 | 58.17 | 59.07 | 55.60 | 55.48 | 54.49 | 53.77 | 53.78 | 63.31 |
| 25% | DER&Fix+USP(Duan et al., 2025) | 80.40 | 76.60 | 79.87 | 79.20 | 71.76 | 66.67 | 64.57 | 60.80 | 58.18 | 56.40 | 55.89 | 58.70 | 57.66 | 58.57 | 55.39 | 53.42 | 51.04 | 54.69 | 56.82 | 55.54 | 62.61 |
| | iCaRL&Fix+**DiL** (Ours) | 87.30 | 86.40 | 85.27 | **85.75** | 78.76 | 76.40 | 73.97 | 71.62 | 70.01 | 66.97 | 64.13 | 67.35 | 66.23 | 66.19 | 65.49 | 66.53 | 65.64 | 65.41 | 65.17 | 64.93 | 71.97 |
| | DER&Fix+**DiL** (Ours) | **88.80** | **87.15** | **86.37** | 84.42 | **83.20** | **81.42** | **78.53** | **77.99** | **77.29** | **75.15** | **74.41** | **71.97** | **69.95** | **68.42** | **68.02** | **67.01** | **67.34** | **66.17** | **65.32** | **66.89** | **75.34** |

*Table 4.* Performance comparison on the 10-task ImageNet-100 benchmark under two different semi-supervised settings. DiL consistently achieves superior performance across both settings.

| Method | ImageNet100-13 | | ImageNet100-100 | |
|---|---|---|---|---|
| | Avg | Last | Avg | Last |
| iCaRL (Rebuffi et al., 2017) | 19.89 | 12.88 | 30.78 | 16.68 |
| NNCSL (Kang et al., 2023) | 42.19 | 33.64 | 56.78 | 53.84 |
| iCaRL&Fix (Fan et al., 2024) | 26.37 | 15.58 | 37.49 | 21.02 |
| +DSGD (Fan et al., 2024) | 28.35 | 19.14 | 50.53 | 32.10 |
| +USP (Duan et al., 2025) | 43.91 | 35.40 | 56.84 | 50.36 |
| **+DiL (Ours)** | **63.24** | **47.44** | **74.90** | **64.74** |
| DER&Fix (Fan et al., 2024) | 35.40 | 29.22 | 61.96 | 52.91 |
| +DSGD (Fan et al., 2024) | 35.73 | 31.53 | 62.27 | 52.82 |
| +USP (Duan et al., 2025) | 46.09 | 39.58 | 62.29 | 55.01 |
| **+DiL (Ours)** | **67.03** | **54.40** | 74.66 | **66.19** |

*Table 5.* Ablation study on the main components of the proposed framework, conducted on the 5-task CIFAR10-30 benchmark.

| Method | Component | | | Avg | Last |
|---|---|---|---|---|---|
| | Distill. | CACE | Mconf | | |
| iCaRL&Fix | | | | 44.02 | 30.71 |
| +CACE+Mconf | | ✓ | ✓ | 76.61 | 64.96 |
| +DPD | ✓ | | | 74.82 | 64.78 |
| +DCD | ✓ | | | 77.15 | 68.2 |
| +DPD+CACE | ✓ | ✓ | | 75.41 | 65.61 |
| +DCD+CACE | ✓ | ✓ | | 77.83 | 69.26 |
| +DPD+CACE+Mconf | ✓ | ✓ | ✓ | 76.39 | 66.74 |
| +DiL | ✓ | ✓ | ✓ | **79.82** | **71.55** |

(Fig. 6 (b)). Accordingly, the feature center of class $c$ after training on task $t$ is denoted as $\mu_{t,c}$ and defined as $\mu_{t,c} = \frac{1}{|D_c|} \sum_{(x,y) \in D_c} f_t(x)$. Based on these centers, the center drift is defined as $\mathrm{CD}_t = \frac{1}{|C_{1:t}|} \sum_{c \in C_{1:t}} \|\mu_{t,c} - \mu_{t-1,c}\|_2$, where $C_{1:t}$ denotes the set of classes observed up to task $t$.

**Baselines.** We mainly compare DiL with recent state-of-the-art SSCL methods, including USP (Duan et al., 2025), DSGD (Fan et al., 2024), under the same experimental settings. To provide a broader reference, we also include representative baselines with complementary design choices, such as NNCSL (Kang et al., 2023), ORDisCo (Wang et al., 2021), and CCIC (Boschini et al., 2022a). For fair comparison, all methods are instantiated on the same SSCL learners, iCaRL&Fix and DER&Fix (Fan et al., 2024).

More implementation details are provided in Appendix B.

### 4.2. Performance comparison

**CIFAR-10 and CIFAR-100.** Table 1 summarizes *Avg* and *Last* accuracy on CIFAR-10/100 under two standard SS-CIL protocols. DiL consistently outperforms strong baselines such as DSGD and USP across datasets and settings, with more pronounced gains on CIFAR-100 where longer task horizons intensify confusion. Notably, compared with the

state-of-the-art baseline USP, DiL improves *Last* accuracy by 5.54% on CIFAR100-20 and 5.04% on CIFAR100-25 in low-label CIFAR-100 settings. These results indicate that DiL is more robust than most existing methods, especially under longer task horizons and severe class confusion.

**CUB.** The results on CUB are shown in Table 2. We evaluate DiL on the 11-task few-shot CUB-200 benchmark under the UaD-CIE protocol. An average base-novel accuracy gap of 3.18% is observed under DiL, while the corresponding average gaps under USP and UaD-CIE are both above 10% in this setting. Moreover, DiL achieves the best last-session accuracy for the novel groups introduced at each task, indicating stronger retention of newly learned classes. This reduced gap on CUB suggests a more balanced performance across classes, indicating that DiL improves stability in learning novel classes while preserving prior knowledge.

**ImageNet-100.** Table 3 shows results under the harder 20-task ImageNet-100 SSCL protocol with varying labeled ratios. DiL achieves substantial improvements in *Avg* accuracy over the state-of-the-art USP in both high- and low-label regimes, with gains of 12.03% under 25% labels and 14.85% under 1% labels. Further experiments are conducted on 10-task ImageNet-100 with different task configurations, where DiL achieves the best overall performance under both frameworks in Table 4. The results show DiL maintains leading performance throughout training, and the advantage suggests robustness to longer horizons and shifts.

*Table 6.* Performance comparison between USP and DiL across different buffer sizes.

| Buffer Size | USP | | DiL | |
|---|---|---|---|---|
| | Avg | Last | Avg | Last |
| 250 | 71.66 | 59.93 | 76.58 | 65.06 |
| 500 | 73.21 | 61.75 | 78.21 | 69.23 |
| 5120 | 79.66 | 70.43 | 79.82 | 71.55 |

*Table 7.* Performance comparison with semi-supervised continual learning baselines under a large buffer regime.

| Method (Buffer Size = 20K) | CIFAR100-125 | | ImageNet100-100 | |
|---|---|---|---|---|
| | Avg | Last | Avg | Last |
| iCaRL & Fix | 62.07 | 46.56 | 40.40 | 26.91 |
| +USP | 68.65 | 55.17 | 56.91 | 51.73 |
| +DiL | 67.08 | 54.04 | 76.59 | 68.06 |
| DER & Fix | 68.75 | 54.83 | 62.02 | 53.46 |
| +USP | 70.60 | 61.33 | 62.17 | 58.34 |
| +DiL | **71.94** | **62.52** | **78.37** | **70.14** |

Beyond the above results, the supplementary material provides extended evaluations on CUB-200 (Section C.1) and robustness analyses under data imbalance (Section C.2).

**Component Ablation.** To quantify the contribution of each component, Table 5 summarizes the comparative ablation results on the 5-task CIFAR10-30 benchmark. In this table, CACE refers only to the anchor-based representation module and does not include confusion-matrix-guided assignment. The confusion matrix $M_{\text{conf}}$ (Sec. 3.2) mitigates interference from confusable base classes by guiding block assignment in CACE when novel classes are introduced. To analyze selective distillation, we introduce Direct Prototype Distillation (DPD), which directly distills full prototypes as a control baseline, in contrast to DCD that selectively distills discriminative dimensions. Comparing DPD and DCD shows that selective distillation is substantially more effective. Specifically, replacing DCD with DPD causes the *Last* accuracy to drop from 68.20% to 64.78%. Removing $M_{\text{conf}}$ further decreases *Last* accuracy by 2.29%, demonstrating that confusion-guided assignment contributes to preserving late-stage performance by improving class separation. These results verify that DCD and CACE are complementary, with the former stabilizing learning and the latter improving class separation under class expansion.

**Memory Buffer Study.** Table 6 evaluates performance across varying exemplar buffer sizes. While the main text focus on standard (5120) budgets, here we investigate restricted scenarios. DiL consistently maintains a significant margin over USP even under severe memory constraints, proving its gains are not dependent on large replay sets. Furthermore, as shown in Table 7, DiL continues to surpass USP under the 20K regime, particularly on ImageNet100-100 using both iCaRL&Fix and DER&Fix backbones.

Additional ablations on architectural choices and key hyper-

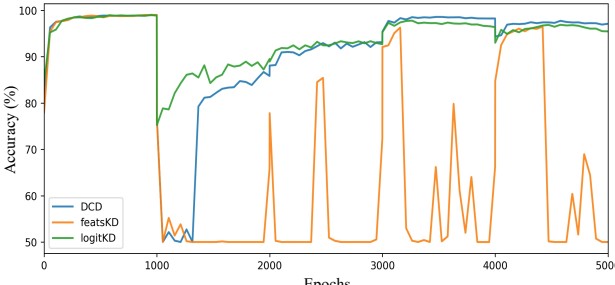

*Figure 5.* Evolution of accuracy on unlabeled data over training epochs at finer epoch granularity. We compare DCD against the traditional featsKD and logitKD distillation methods.

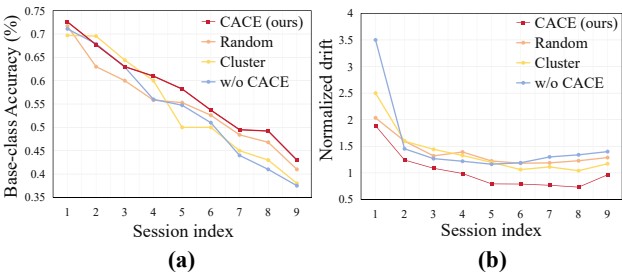

*Figure 6.* Effect of CACE on stability and drift. (a) Base-class accuracy across tasks comparing CACE with other prototype initialization strategies. (b) Center drift computed from class centers, where lower values indicate more stable representations.

parameters are provided in detail in Appendix C.4.

### 4.3. Analysis of Design Effectiveness

**Effectiveness of DCD.** To evaluate the role of DCD in suppressing error accumulation, Fig. 5 compares it with other distillation methods on the 5-task CIFAR10-30 setting. As shown by the blue curve, DCD maintains higher late-stage accuracy and recovers quickly after each task transition. In contrast, FeatKD suffers from drastic oscillations and repeated collapses to low accuracy, while LogitKD is more stable but achieves lower late-stage accuracy than DCD. The curve trends indicate that our distillation mitigates noise-induced degradation, especially in later training stages.

**Effectiveness of CACE.** We conduct additional analysis to assess the contribution of sparse block encoding under the 10-task CIFAR100-20 setting. Fig. 6 (a) shows that CACE maintains higher base-class accuracy than random or clustering-based initialization across sessions, indicating stronger retention when novel classes are introduced. Moreover, removing CACE leads to a larger accuracy drop, highlighting the role of structured encoding in stabilizing base classes. Fig. 6 (b) further shows lower normalized drift across most sessions, indicating a more reliable neighborhood structure. In contrast, random or clustering-based initialization yields higher drift, and the drift increases fur-

*Table 8.* Task-wise comparison of Relative Drift (RelDrift) and Boundary Margin (BM) across different methods.

| Method | Metric | Task ID | | | | | | | | | | Avg |
|---|---|---|---|---|---|---|---|---|---|---|---|---|
| | | 1 | 2 | 3 | 4 | 5 | 6 | 7 | 8 | 9 | 10 | |
| iCaRL | RelDrift | 0 | 1.0187 | 1.0974 | 1.1769 | 1.2386 | 1.3018 | 1.3575 | 1.4129 | 1.4638 | 1.5196 | 1.1587 |
| | BM | 0.3618 | 0.0927 | 0.0134 | -0.0228 | -0.0539 | -0.0794 | -0.1036 | -0.1261 | -0.1457 | -0.1669 | -0.0231 |
| DER | RelDrift | 0 | 0.9473 | 1.0068 | 1.0661 | 1.1235 | 1.1684 | 1.2149 | 1.2531 | 1.2917 | 1.3286 | 1.0400 |
| | BM | 0.4026 | 0.1439 | 0.0786 | 0.0237 | -0.0084 | -0.0321 | -0.0537 | -0.0704 | -0.0869 | -0.0998 | 0.0298 |
| NNCSL | RelDrift | 0 | 0.7736 | 0.7403 | 0.8354 | 0.8755 | 0.8894 | 0.9456 | 1.0241 | 1.0544 | 1.1059 | 0.8244 |
| | BM | 0.4729 | 0.1837 | 0.1176 | 0.0738 | 0.0479 | 0.0271 | 0.0094 | -0.0035 | -0.0164 | -0.0298 | 0.0883 |
| DSGD | RelDrift | 0 | 0.9004 | 0.8388 | 0.8961 | 0.9657 | 0.9946 | 1.1311 | 1.1146 | 1.1692 | 1.1702 | 0.9181 |
| | BM | 0.4484 | 0.1416 | 0.0984 | 0.0608 | 0.0403 | 0.0329 | 0.0137 | -0.0053 | -0.0193 | -0.0385 | 0.0773 |
| USP | RelDrift | 0 | 0.8465 | 0.8241 | 0.8768 | 0.9326 | 0.9817 | 1.0234 | 1.0689 | 1.1082 | 1.1476 | 0.8810 |
| | BM | 0.5004 | 0.1924 | 0.1138 | 0.0522 | 0.0318 | 0.0223 | 0.007 | -0.0141 | -0.0246 | -0.0454 | 0.0836 |
| **DiL** | RelDrift | 0 | 0.7824 | 0.7589 | 0.8036 | 0.8427 | 0.8794 | 0.9138 | 0.9431 | 0.9715 | 1.0028 | 0.7898 |
| | BM | 0.4917 | 0.2146 | 0.1478 | 0.1089 | 0.0796 | 0.0573 | 0.0378 | 0.0196 | 0.0068 | -0.0051 | 0.1159 |

*Table 9.* Task-wise average pairwise cosine similarity among class anchors on CIFAR-10, CIFAR-100, and CUB-200.

| Dataset | Task ID | | | | | | | | | | |
|---|---|---|---|---|---|---|---|---|---|---|---|
| | 1 | 2 | 3 | 4 | 5 | 6 | 7 | 8 | 9 | 10 | 11 |
| CIFAR10 | 0 | 0 | 0 | 0 | 0 | - | - | - | - | - | - |
| CIFAR100 | 0.0431 | 0.0337 | 0.0328 | 0.0280 | 0.0240 | 0.0214 | 0.0194 | 0.0183 | 0.0177 | 0.0173 | - |
| CUB200 | 0.0334 | 0.0317 | 0.0308 | 0.0302 | 0.0296 | 0.0290 | 0.0287 | 0.0283 | 0.0262 | 0.0258 | 0.0216 |

ther without CACE.

**Analysis of Feature Drift and Boundary Margin.** To directly validate how our framework mitigates feature drift and boundary erosion, we introduce two metrics: Relative Drift (RelDrift) and Boundary Margin (BM). RelDrift extends the drift analysis in Fig. 6 (b) to a cross-method comparison. For a test sample $x$ with ground-truth label $y$ and extracted feature $f_t(x)$, its nearest competing class is defined as $r_t(x) = \arg\min_{c \neq y} d\left(f_t(x), \mu_c^{(t)}\right)$, and the corresponding boundary margin is calculated as $m_t(x) = d\left(f_t(x), \mu_{r_t(x)}^{(t)}\right) - d\left(f_t(x), \mu_y^{(t)}\right)$. As shown in Table 8, on the 10-task CIFAR100-20 benchmark, DiL achieves the lowest average RelDrift (0.7898) and the highest average BM (0.1159).

**Analysis of Inter-Class Orthogonality.** To assess the inter-class orthogonality induced by our framework, we compute the average pairwise cosine similarity among class anchors across sequential tasks. We follow the standard task protocols (i.e., 5, 10, and 11 tasks for CIFAR-10, CIFAR-100, and CUB-200, respectively) and the block configurations detailed in Table 10. As shown in Table 9, the average similarity quickly drops below 0.05 and continues approaching 0, which supports the near-orthogonal effect of CACE.

## 5. Conclusion

We presented DiL, a discrete-anchored framework for semi-supervised continual learning that addresses pseudo-label noise accumulation and class confusion through structured representation alignment. DiL integrates two components:

(1) Discrete Contrastive Distillation (DCD), which performs selective distillation on informative channels to suppress noise propagation; and (2) Class-Aware Channel-Chunked Encoding (CACE), which assigns block-sparse anchors to enlarge class margins and separate novel classes from confusable base classes. Extensive experiments across SSCL protocols show that DiL improves performance and stability under limited supervision. Despite these gains, fixed channel selection and anchor configurations may become less effective over time. In future work, we will explore adaptive channel and anchor design under changing data streams.

## Acknowledgements

This research was supported by the National Natural Science Foundation of China (No. 62506089) , Scientific and Technological Innovation Platform Research Project of Guizhou Province (CXPTXM[2025]024), Guizhou Province Youth Science and Technology Talent Project ([2024]317) , Guizhou Provincial Science and Technology Projects ([2024]002, CXTD [2023]027), Guizhou Provincial Basic Research Program (Natural Science) (Qiankehe Foundation MS[2026]080).

## Impact Statement

This paper presents work whose goal is to advance the field of Semi-supervised Learning. There are many potential societal consequences of our work, none which we feel must be specifically highlighted here.

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

# A. Feasible Set Derivation for $\mathcal{F}$

We build block-sparse class anchors by partitioning the $d$-dimensional feature space into $B$ disjoint chunks of size $d/B$ and assigning each class a $k$-chunk support set $S \in \mathcal{E}$. Since the activated chunks do not overlap, each anchor lies in a subspace of dimension equal to the activated coordinates. We define the effective dimension as:

$$d_{\text{eff}} = k\frac{d}{B}. \tag{16}$$

Intuitively, $d_{\text{eff}}$ should be (i) large enough to preserve the relative geometry among $C$ class anchors, yet (ii) small enough to keep the encoding sparse and computationally efficient. We formalize these requirements as three constraints.

## A.1. Capacity constraint

Recall that the $k$-block pattern set is defined as $\mathcal{E}$ in Eq. (11), hence $|\mathcal{E}| = \binom{B}{k}$. Assigning distinct patterns to $C$ classes requires:

$$\binom{B}{k} \geq C. \tag{17}$$

This ensures class-level identifiability, as each class is assigned a unique active-block pattern.

## A.2. JL-type dimension requirement

Beyond ensuring a sufficient number of distinct chunk supports, we require that the anchor geometry is preserved under the projection onto the selected-chunk subspace. A Johnson-Lindenstrauss (JL)-type (Johnson & Lindenstrauss, 1984)result states that, for a set of $C$ points, there exists a mapping into $\mathbb{R}^m$ that approximately preserves all pairwise distances up to distortion $\varepsilon \in (0, 1)$, provided:

$$m \geq \kappa\frac{\log C}{\varepsilon^2}, \tag{18}$$

where $\kappa > 0$ is a universal constant. In our construction, class anchors are embedded in the activated subspace of dimension $d_{\text{eff}}$, so we impose $d_{\text{eff}} \geq m$. Using Eq. (16), this yields:

$$k\frac{d}{B} \geq \kappa\frac{\log C}{\varepsilon^2}. \tag{19}$$

This constraint prevents the anchor space from being too low-dimensional to maintain inter-class separation.

## A.3. Sparsity budget

To control redundancy and computation, we enforce a maximum activation ratio $\gamma_{\max} \in (0, 1)$. This yields the sparsity budget:

$$d_{\text{eff}} \leq \gamma_{\max}d. \tag{20}$$

By Eq. (16), this is equivalent to $\frac{k}{B} \leq \gamma_{\max}$, which limits the fraction of activated chunks and keeps the encoding

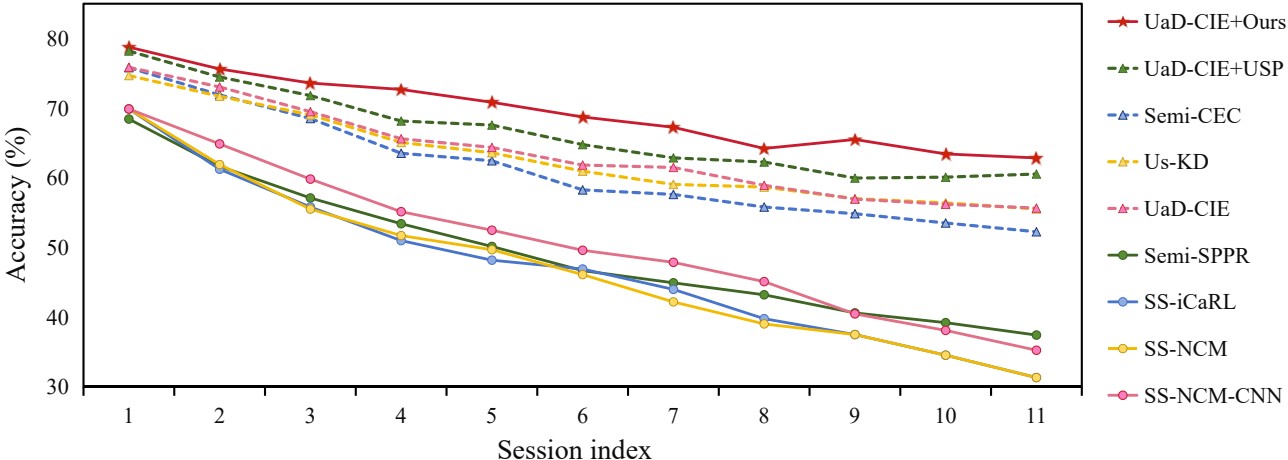

*Figure 7.* Comparison of average accuracy across 11 incremental tasks on the CUB-200 benchmark. This evaluation incorporates a broader set of baselines, illustrating the evolution of overall accuracy over all learned classes at each stage.

block-sparse. Combining Eq. (17), Eq. (19), and Eq. (20) yields the feasible parameter domain $\mathcal{F}$ in Eq. (21). These design constraints define the feasible domain:

$$\mathcal{F} = \left\{ (B,k) \in \mathbb{N}^2 \ \middle| \ \begin{array}{l} \binom{B}{k} \geq C, \\[6pt] \kappa \dfrac{\log C}{\epsilon^2} \leq \dfrac{k}{B}d \leq \gamma_{\max}d \end{array} \right\}. \quad (21)$$

### A.4. Practical instantiation of $(B,k)$ for training

To make the feasible-set derivation actionable, we instantiate $(B,k)$ by a short discrete search.

**Search space.** We fix the feature dimension to $d = 512$ based on the sensitivity study in Table 16, and restrict $B$ to a small candidate set $\mathcal{B}$ such that $B \mid d$ (e.g., $\mathcal{B} = \{8, 16, 32, 64\}$). For each $B \in \mathcal{B}$, we enumerate $k \in \{1, \ldots, \lfloor \gamma_{\max} B \rfloor\}$.

**Constraints.** For each candidate $(B,k)$, we test feasibility by: (i) capacity $\binom{B}{k} \geq C$ (Eq. (17)), (ii) JL-type requirement $k\frac{d}{B} \geq \eta \log C$ (Eq. (19)), and (iii) sparsity budget $\frac{k}{B} \leq \gamma_{\max}$ (Eq. (20)), where $\eta = \kappa/\varepsilon^2$ absorbs JL constants into a single coefficient.

**Selection rule.** Among all feasible pairs, we select a fixed pair $(B^*, k^*)$ within $\mathcal{F}$ by minimizing pattern-set redundancy:

$$(B^*, k^*) = \arg \min_{(B,k) \in \mathcal{F}} \frac{\binom{B}{k} - C}{C}. \quad (22)$$

When multiple candidates yield similar redundancy, we prefer a smaller $B$ for efficiency. In our experiments, we adopt the $(B^*, k^*)$ settings listed in Table 10 for representative class counts.

*Table 10.* Example instantiations of $(B,k)$ from the feasible set $\mathcal{F}$ with $d = 512$. The pair $(B^*, k^*)$ is selected by redundancy minimization.

| $C$ | $B^*$ | $k^*$ | $\binom{B^*}{k^*}$ | $d/B^*$ | $d_{\text{eff}} = k^*d/B^*$ |
|-----|-------|-------|--------------------|---------|------------------------------|
| 10  | 10    | 1     | 10                 | 51.20   | 51.20                        |
| 100 | 15    | 2     | 105                | 34.13   | 68.27                        |
| 200 | 12    | 3     | 220                | 42.67   | 128.00                       |

For the first task, we initialize the class chunk supports by sequential chunk allocation. Newly introduced classes in later tasks then follow the above allocation rule.

## B. Implementation Details

### B.1. Training Configuration

ResNet-32 (He et al., 2016) is used for CIFAR-10 and CIFAR-100, while ResNet-18 (He et al., 2016) is adopted for ImageNet-100 and CUB. For unlabeled data, we follow FixMatch (Sohn et al., 2020) with weak–strong augmentations and confidence-based pseudo-label filtering using $\tau = 0.95$. We adopt the same optimizer and basic training configuration as USP (Duan et al., 2025), including SGD-based optimization with momentum and weight decay, a 10-epoch warm-up, and a cosine-annealing learning-rate schedule for 200 epochs.

### B.2. Detailed Algorithmic Workflow

To explicitly enhance the algorithmic clarity and guarantee the reproducibility of our proposed framework, we present the comprehensive training pipeline of Discrete-anchored Incremental Learning (DiL) at task $t$ in Algorithm 1. This complete workflow complements the conceptual overview

**Algorithm 1** DiL Training Procedure at Task $t$

**Require:** Labeled set $D_l^t$, unlabeled stream $D_u^t$, replay buffer $E^t$, frozen teacher $\theta_{t-1}$

**Ensure:** Updated student model $\theta_t$

1: Initialize student $\theta_t \leftarrow \theta_{t-1}$ and freeze teacher $\theta_{t-1}$
2: // *Stage 1: Discrete Prototype and Channel Selection*
3: **for** all class $c \in \mathcal{Y}_{all}^t$ **do do**
4:   $P_c \leftarrow$ Compute class prototype mean over labeled samples in $D_{l,c}^t$
5:   $M_c \leftarrow$ Evaluate channel importance scores via magnitude and frequency
6:   $\Omega_c \leftarrow$ Extract $K_{top}$ channel indices from $M_c$
7:   $\hat{P}_c \leftarrow$ Construct normalized discrete anchor prototype using binary mask $m_c$ from $\Omega_c$
8: **end for**
9: // *Stage 2: Confusion-Aware Subspace Partitioning*
10: Construct block partition space $\mathcal{E}$ with block-sparse chunk supports
11: **for** all novel class $n \in \mathcal{Y}_n^t$ **do do**
12:   Identify confusable base classes to form confusion set $\mathcal{N}_{conf}(n)$ via $\langle \hat{P}_n, \hat{P}_c \rangle$
13:   $S_n \leftarrow$ Determine optimal block-sparse chunk assignment by minimizing overlap with $\mathcal{N}_{conf}(n)$ over $\mathcal{E}$
14: **end for**
15: // *Stage 3: Incremental Representation Learning*
16: **while** not converged **do do**
17:   Compute standard semi-supervised losses $\mathcal{L}_{sup}$ and $\mathcal{L}_{uns}$ on labeled and unlabeled samples
18:   $\mathcal{L}_{cl} \leftarrow$ Apply experience replay distillation on buffer $E^t$
19:   $\mathcal{L}_{DCD} \leftarrow$ Perform partitioned distillation on masked prototype-aligned representations
20:   $\mathcal{L}_{CACE} \leftarrow$ Compute anchor-guided encoding loss between novel features and anchors $S_n$
21:   Update student parameters $\theta_t$ by minimizing total objective $\mathcal{L}_{total}$
22: **end while**
23: **return** Updated model $\theta_t$

presented in Section 3.2 of the main text by meticulously illustrating the execution sequence, data flow, and optimization mechanics across individual tasks.

## B.3. Replay and Memory

Following USP, we maintain an exemplar buffer with a fixed total budget $M$. After each task, when $k$ classes have been observed, we allocate $m_t = \lceil M/k \rceil$ exemplars per class. Exemplars for novel classes are selected using an iCaRL-style herding procedure to approximate the class-mean feature, and base-class exemplar sets are reduced by retaining the first $m_t$ samples. During training, replay exemplars are mixed with current-task mini-batches, and replay distillation is applied on the buffer to preserve previously

*Table 11.* Average and Last accuracy on the CIFAR 100-20 semi-supervised benchmark evaluated under class imbalance and data scale imbalance scenarios.

| Method | Longtail | | Taskimb | |
|---|---|---|---|---|
| | Avg | Last | Avg | Last |
| USP | 55.01 | 40.47 | 54.81 | 39.99 |
| **DiL** | **55.70** | **44.01** | **56.92** | **43.35** |

*Table 12.* Comparison of computational efficiency and resource overhead across different methods.

| Method | Params | Peak Memory | Training Time | FLOPs |
|---|---|---|---|---|
| UaD-CIE | 372000 | 4022 MB | 120 min/task | 40 G |
| NNCSL | 868781 | 3200 MB | 100.06 min/task | 55.02 G |
| DSGD | 528969 | 716 MB | 65 min/task | 39.5 G |
| USP | 539316 | 4949 MB | 85.02 min/task | 35.45 G |
| DIL | 539316 | 3825 MB | 102.94 min/task | 35.45 G |

learned knowledge.

## B.4. Module Hyperparameters

We set the feature dimension to 512 and the memory budget to 5120 based on ablation studies, and set all loss weights $\lambda_{uns}$, $\lambda_{cl}$, $\lambda_{dcd}$, and $\lambda_{cace}$ to 1. The encoder output is projected by a lightweight projection head whose output dimension is 512, matching the feature dimension used in both DCD and CACE. Replay-based distillation applies temperature scaling with $\tau_{kd} = 0.1$. In DCD, we use a shared Top-$K_{top}$ for both class- and sample-level selection to keep the sparsity strength consistent. $K_{top}$ is chosen via the sensitivity study in Fig. 8 (b) across benchmarks of different class cardinalities, then fixed for all datasets. In CACE, the confusion set size is set to $K_{conf} = 1$ by default for semantic-avoidance assignment, and $(B, k)$ is determined from the feasible constraints in Eq. (22).

## C. Supplementary Experiments.

### C.1. Broader Baseline Comparison on CUB-200

In addition, we compare against other representative approaches that exploit unlabeled data via different mechanisms, including Semi-CEC (Cui et al., 2021), Us-KD (Cui et al., 2022), and Semi-SPPR (Zhu et al., 2021), as well as replay-based Semi-supervised Continual Learning baselines SS-iCaRL (Cui et al., 2021), SS-NCM (Cui et al., 2021), and SS-NCM-CNN (Cui et al., 2021). All methods follow

*Table 13.* Impact of different backbone architectures on performance. We compare our method against the USP baseline across ResNet-20, ResNet-32, and ResNet-50.

| Backbone | ResNet20 | | ResNet32 | | ResNet50 | |
|---|---|---|---|---|---|---|
| | Avg | Last | Avg | Last | Avg | Last |
| USP | 51.34 | 40.97 | 53.20 | 41.30 | 33.42 | 27.29 |
| **DiL** | **55.69** | **42.94** | **56.16** | **45.55** | **38.40** | **28.51** |

*Table 14.* Performance comparison using pre-trained Vision Transformer backbones (CLIP-ViT and DINO-ViT) on CIFAR10-30.

| Method | CLIP | | DINO | |
|---|---|---|---|---|
| | Avg | Last | Avg | Last |
| DSGD | 74.83 | 73.15 | 78.61 | 72.43 |
| USP | 81.64 | 75.42 | 80.95 | 74.21 |
| DiL | **82.27** | **76.28** | **81.46** | **75.19** |

*Table 15.* Sensitivity analysis of the pseudo-label confidence threshold on the 5-task CIFAR10-30.

| Threshold | 0.5 | 0.6 | 0.7 | 0.8 | 0.9 | 0.95 | 0.99 |
|---|---|---|---|---|---|---|---|
| Avg | 59.04 | 52.22 | 59.08 | 63.92 | 72.51 | **79.82** | 78.14 |
| Last | 48.44 | 58.84 | 60.53 | 61.70 | 66.07 | **71.55** | 69.57 |

the same 11-task CUB-200 setting as in Table 2. Fig. 7 plots overall accuracy over 11 tasks, and DiL maintains a consistently higher trajectory. The advantage remains stable in later tasks, where competing methods typically degrade more, which is consistent with the strong Novel gains.

## C.2. Robustness under Data Imbalance

Robustness is assessed under two imbalance regimes, including long-tailed class imbalance and task-wise data quantity imbalance. In the long-tailed setting, each session keeps a similar overall data budget, but the 10 within-session classes follow a long-tailed distribution with imb_factor=50, so the challenge mainly comes from intra-session class-frequency skew. In the task-imbalance setting, class balance within each session is not explicitly skewed, while the session data budget and unlabeled ratio progressively decrease, making later sessions increasingly data-scarce. Table 11 shows that DiL consistently surpasses USP in both settings, particularly in *Last* accuracy.

## C.3. Computational Efficiency Analysis

To evaluate the computational overhead, we conduct unified efficiency experiments with all methods instantiated on the iCaRL&Fix baseline under identical hardware and backbone settings. As shown in Table 12, DiL introduces no extra trainable parameters or inference FLOPs compared to the SOTA method USP, while successfully reducing peak memory utilization (e.g., from 4949 MB to 3825 MB on CIFAR100-20). Although explicitly addressing both feature drift and boundary erosion incurs a moderate increase in training time, this controlled complexity is highly justified.

*Table 16.* Sensitivity analysis of the feature dimension size on the 5-task CIFAR10-30.

| Dimensions | 64 | 128 | 256 | 512 | 1024 | 2408 |
|---|---|---|---|---|---|---|
| Avg | 75.80 | 78.40 | 76.20 | **79.82** | 78.15 | 77.32 |
| Last | 68.80 | 71.05 | 69.25 | **71.55** | 71.13 | 69.27 |

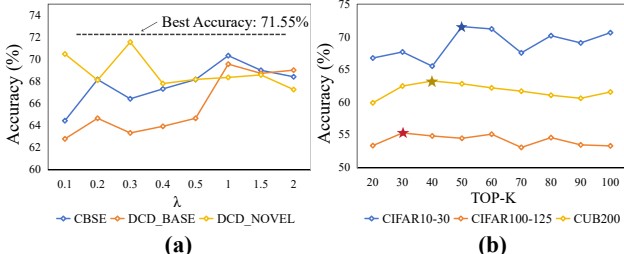

*Figure 8.* Hyperparameter sensitivity analysis. (a) Ablation study on loss weights $\lambda$ for CACE, DCD_BASE, and DCD_NOVEL on CIFAR10-30, illustrating the final classification accuracy under different weight configurations. (b) Evaluation of average accuracy with varying $K_{\text{top}}$ values across datasets of different scales.

*Table 17.* Sensitivity analysis of the weight parameters $\lambda_{\text{uns}}$ and $\lambda_{\text{cl}}$.

| Weight | Metric | 0.1 | 0.5 | 1 | 1.5 | 2 |
|---|---|---|---|---|---|---|
| $\lambda_{\text{uns}}$ | **Avg** | 76.88 | 78.94 | **79.82** | 78.21 | 77.36 |
| | **Last** | 66.92 | 69.48 | **71.55** | 69.87 | 68.73 |
| $\lambda_{\text{cl}}$ | **Avg** | 73.91 | 77.63 | **79.82** | 78.06 | 76.71 |
| | **Last** | 64.85 | 68.74 | **71.55** | 69.96 | 68.41 |

## C.4. More Ablation Studies

### C.4.1. BACKBONE ROBUSTNESS.

We evaluate different backbones to examine the robustness of DiL to the choice of feature extractor.

**Standard CNNs.** Table 13 first reports performance on ResNet-20/32/50 under the CIFAR100-20 setting, where DiL consistently outperforms USP across all capacities (e.g., achieving 56.16% *Avg* and 45.55% *Last* with ResNet-32).

**Pre-trained ViTs.** To ensure these gains are not limited to CNN structures, we further verify our framework using pre-trained Vision Transformers (CLIP-ViT and DINO-ViT) on CIFAR10-30. As shown in Table 14, DiL remains superior when utilizing both self-supervised (DINO) and vision-language (CLIP) visual encoders.

### C.4.2. HYPERPARAMETER STABILITY.

**Pseudo-label confidence threshold.** Table 15 shows that performance peaks at a threshold of 0.95 (79.82% average

*Table 18.* Sensitivity analysis of the confusion set size $K_{conf}$ on CIFAR100-20 and CUB-200.

| $K_{\text{conf}}$ | CIFAR100-20 | | cub200 | |
|---|---|---|---|---|
| | Avg | Last | Avg | Last |
| 1 | 56.16 | 45.55 | **69.32** | **61.86** |
| 2 | **57.92** | **46.18** | 69.18 | 61.62 |
| 3 | 56.61 | 45.97 | 69.01 | 61.28 |
| 5 | 54.88 | 44.12 | 67.94 | 59.83 |
| 7 | 53.36 | 42.71 | 66.38 | 58.07 |

*Table 19.* Quantitative ablation study on the block configurations $(B, k)$ across different datasets.

| Dataset | $(B, K)$ | Avg | Last |
|---|---|---|---|
| CIFAR10-30 | $(6, 1)$ | 72.96 | 63.90 |
| | $(8, 1)$ | 68.35 | 63.36 |
| | $(10, 1)$ | **79.82** | **71.55** |
| | $(12, 1)$ | 78.17 | 70.58 |
| | $(16, 1)$ | 78.37 | 70.42 |
| | $(10, 2)$ | 76.09 | 69.35 |
| | $(10, 3)$ | 74.54 | 65.24 |
| | $(10, 4)$ | 70.23 | 62.34 |
| CIFAR100-20 | $(10, 2)$ | 51.33 | 33.61 |
| | $(12, 2)$ | 48.38 | 42.50 |
| | $(15, 2)$ | 56.16 | **45.55** |
| | $(18, 2)$ | 55.21 | 44.02 |
| | $(20, 2)$ | 45.22 | 43.42 |
| | $(15, 1)$ | 51.76 | 33.35 |
| | $(15, 3)$ | **56.17** | 42.58 |
| | $(15, 4)$ | 55.54 | 45.38 |
| CUB200 | $(8, 3)$ | 58.74 | 51.62 |
| | $(10, 3)$ | 64.85 | 57.88 |
| | $(12, 3)$ | **69.32** | **61.86** |
| | $(14, 3)$ | 68.71 | 61.42 |
| | $(16, 3)$ | 68.10 | 60.94 |
| | $(12, 2)$ | 62.14 | 54.83 |
| | $(12, 4)$ | 68.43 | 61.77 |
| | $(12, 5)$ | 66.91 | 59.28 |

accuracy). Further increasing it to 0.99 causes a slight drop, indicating that an overly strict threshold limits the availability of usable pseudo-labels.

**Feature dimensionality.** Table 16 investigates the influence of the feature dimensionality, showing a relatively flat trend. The best performance is achieved at 512 dimensions with 79.82% *Avg* and 71.55% *Last*, while larger dimensions introduce only minor fluctuations, suggesting an overall low sensitivity to the representation size.

**Loss weights.** Fig. 8 (a) and Table 17 present the sensitivity to various loss weights. In Fig. 8 (a), the final accuracy changes smoothly under different weights for the DCD and CACE components. Table 17 evaluates the global unsupervised weight $\lambda_{\mathrm{uns}}$ and the replay distillation weight $\lambda_{\mathrm{cl}}$.

**Top-K Selection Parameters.** Fig. 8 (b) shows that the average accuracy remains stable across different $K_{\mathrm{top}}$, indicating low sensitivity to this parameter. For $K_{\mathrm{conf}}$ (Table 18), CUB-200 performs best at $K_{\mathrm{conf}} = 1$, while CIFAR100-20 peaks slightly at $K_{\mathrm{conf}} = 2$.

### C.4.3. DESIGN VALIDATION.

**Block configurations.** To empirically validate the feasible set derivation discussed in Table 10, Table 19 presents quantitative ablations on the block configurations $(B, k)$ across CIFAR10-30, CIFAR100-20, and CUB-200. We evaluate two protocols: varying total blocks $B$ while fixing active blocks $k$, and vice versa. The results demonstrate that the theoretically derived configurations are optimal or near-optimal across all datasets.

