# OpenReview forum: "DiL: Discrete-anchored Representation Alignment for Semi-Supervised Continual Learning"
_ICML.cc/2026/Conference — ICML 2026 regular_

### Official Review · Reviewer_TMhQ · 2026-02-27

**Soundness:** 3
**Presentation:** 3
**Significance:** 3
**Originality:** 3
**Overall Recommendation:** 4
**Confidence:** 4

**Summary:**

This paper studies semi-supervised class incremental learning and proposes a framework called discrete anchored incremental learning (DiL). DiL consists of two key components. First, discrete contrastive distillation (DCD) achieves selective distillation by identifying class-related feature channels. Second, class-aware channel chunked encoding divides the feature space into blocks and assigns sparse block support to each class. For new classes, block assignment is based on their similarity to existing base classes, thus promoting the separation of the representation space.

**Compliance With Llm Reviewing Policy:**

Affirmed.

**Final Justification:**

The author addressed my concerns, and I will raise my rating.

**Key Questions For Authors:**

Please refer to the strengths and weaknesses.

**Limitations:**

Please refer to the strengths and weaknesses.

**Strengths And Weaknesses:**

Strengths：

1. The method proposed in the paper is  interesting and solves the problem of semi-supervised class incremental learning.

2. The writing is easy to understand, and the illustrations are easy to comprehend.

Weaknesses：

1. DCD performed sparsification on the prototype, the ablation experiments should include a comparison with direct distillation of the prototype to demonstrate its effectiveness.

2. The supplementary materials describe the selection process for B/K, but lack quantitative experimental analysis for different B/K.

3. The motivation behind CACE is to enforce near-orthogonal of inter-class subspaces. While Figure 6 analyzes stability and drift, it does not directly analyze orthogonality.

4. Hyperparameters sensitivity analysis in Eq.(15) should be included.

5. Replay is still needed even though distillation has already been performed with prototype.  Is the method effective when data storage is not required?

If authors can resolve these concerns, I will raising the rating.

---

> ### Author Rebuttal · Authors · 2026-03-31
>
> 1. **Please refer to our response to Reviewer 1’s first question, where we provide DPD-based ablations on CIFAR10-30, CIFAR10-150, and CIFAR100-25.** DPD (Direct Prototype Distillation) denotes the variant that removes the Top-K sparsification step and directly distills the full prototype. The results consistently show that the gain does not come from prototype distillation alone, but mainly from the selective sparsification design of DCD. We will include this comparison explicitly in the revised ablation table.
>
> 2. We thank the reviewer for pointing out this limitation. To directly validate the B/K selection in the appendix, we add quantitative ablations on **CIFAR10-30, CIFAR100-20, and CUB200**, under two protocols: fixing the number of active blocks per class while varying the number of blocks, and vice versa. The results show that the configurations used in the paper are optimal or near-optimal across all three datasets, **consistent with Appendix A**. Too few blocks or too many active blocks increase overlap, whereas overly sparse partitions weaken representation capacity. We will include these results in the revision.
>
> | Dataset | Fixed Setting | Variable | Value | Avg | Last |
> |---|---|---|---|---|---|
> | CIFAR10-30 | K=1 | B | 6 | 72.96 | 63.9 |
> |  | K=1 | B | 8 | 68.35 | 63.36 |
> |  | K=1 | B | 10 | **79.82** | **71.55** |
> |  | K=1 | B | 12 | 78.17 | 70.58 |
> |  | K=1 | B | 16 | 78.37 | 70.42 |
> |  | B=10 | K | 1 | **79.82** | **71.55** |
> |  | B=10 | K | 2 | 76.09 | 69.35 |
> |  | B=10 | K | 3 | 74.54 | 65.24 |
> |  | B=10 | K | 4 | avg | 62.34 |
> | CIFAR100-20 | K=2 | B | 10 | 51.33 | 33.61 |
> |  | K=2 | B | 12 | 48.38 | 42.5 |
> |  | K=2 | B | 15 | 56.16 | **45.55** |
> |  | K=2 | B | 18 | 55.21 | 44.02 |
> |  | K=2 | B | 20 | 45.22 | 43.42 |
> |  | B=15 | K | 1 | 51.76 | 33.35 |
> |  | B=15 | K | 2 | 56.16 | **45.55** |
> |  | B=15 | K | 3 | **56.17** | 42.58 |
> |  | B=15 | K | 4 | 55.54 | 45.38 |
> | CUB200 | K=3 | B | 8 | 58.74 | 51.62 |
> |  | K=3 | B | 10 | 64.85 | 57.88 |
> |  | K=3 | B | 12 | **69.32** | **61.86** |
> |  | K=3 | B | 14 | 68.71 | 61.42 |
> |  | K=3 | B | 16 | 68.1 | 60.94 |
> |  | B=12 | K | 2 | 62.14 | 54.83 |
> |  | B=12 | K | 3 | **69.32** | **61.86** |
> |  | B=12 | K | 4 | 68.43 | 61.77 |
> |  | B=12 | K | 5 | 66.91 | 59.28 |
>
> 3. We thank the reviewer for this helpful suggestion. To directly assess inter-class orthogonality, we compute the average pairwise cosine similarity among class anchors. We follow the corresponding B/K settings in **Table 7** and the standard task protocols in the paper, namely 5, 10, and 11 tasks for **CIFAR-10, CIFAR-100, and CUB-200**, respectively. The average similarity quickly drops below **0.05** and continues approaching **0**, which supports the near-orthogonal effect of CACE. We will include this analysis in the revision.
>
>
> | Dataset | 1 | 2 | 3 | 4 | 5 | 6 | 7 | 8 | 9 | 10 | 11 |
> |---|---|---|---|---|---|---|---|---|---|---|---|
> | CIFAR10 | 0 | 0 | 0 | 0 | 0 | - | - | - | - | - | - |
> | CIFAR100 | 0.0431 | 0.0337 | 0.0328 | 0.028 | 0.024 | 0.0214 | 0.0194 | 0.0183 | 0.0177 | 0.0173 | - |
> | CUB200 | 0.0334 | 0.0317 | 0.0308 | 0.0302 | 0.0296 | 0.029 | 0.0287 | 0.0283 | 0.0262 | 0.0258 | 0.0216 |
>
> 4. We thank the reviewer for this reminder. **Fig. 8(a)** reports ablations for the internal weights $\lambda_{dcd}$ and $\lambda_{cace}$, which show a smooth performance surface. We further add a sensitivity analysis of $\lambda_{uns}$ and $\lambda_{cl}$ on CIFAR10-30, and the results show that DiL remains stable over a wide range, with 1.0 consistently performing best.
>
> | Weight | Metric | 0.1 | 0.5 | 1 | 1.5 | 2 |
> |---|---|---|---|---|---|---|
> | $\lambda_{uns}$ | Avg | 76.88 | 78.94 | **79.82** | 78.21 | 77.36 |
> |  | Last | 66.92 | 69.48 | **71.55** | 69.87 | 68.73 |
> | $\lambda_{cl}$  | Avg | 73.91 | 77.63 | **79.82** | 78.06 | 76.71 |
> |  | Last | 64.85 | 68.74 | **71.55** | 69.96 | 68.41 |
>
> 5. We thank the reviewer for raising this important question. Replay is not an extra assumption of our method, but a standard setting in class-incremental SSCL protocols. **Replay and prototype distillation are also not equivalent**: replay preserves sample-level information to reduce forgetting, while prototype distillation provides class-level constraints to reduce feature drift. To directly test the no-storage setting, we add experiments on CIFAR10-30 with **Buffer Size = 0**, where raw old samples are removed but compact class-level memory is retained. DiL remains clearly effective in this setting, and DCD still outperforms DPD. We will include this experiment in the revised manuscript.
>
> | Method | DPD | DCD | CACE | Avg | Last |
> |---|---|---|---|---|---|
> | iCaRL&Fix | - | - | - | 25.99 | 15.58 |
> | iCaRL&Fix+DPD | Y | - | - | 39.02 | 27.82 |
> | iCaRL&Fix+DCD | - | Y | - | 40.68 | 30.42 |
> | iCaRL&Fix+CACE | - | - | Y | 38.97 | 28.81 |
> | iCaRL&Fix+DPD+CACE | Y | - | Y | 41.8 | 31.32 |
> | iCaRL&Fix+DiL | - | Y | Y | **44.68** | **35.42** |

---

> > ### Author Rebuttal · Reviewer_TMhQ · 2026-04-02
> >
> > I sincerely appreciate the authors’ efforts in addressing the concerns I previously raised. However, I still have reservations regarding the response to Q5. Under the same benchmark, the  performance drops dramatically once the storage of replay data is removed, with the accuracy nearly halved. Particularly the field of CL is increasingly focused on large-scale models, this paper is still working with the ResNet and small-scale datasets such as CIFAR, achieves a classification performance of only around 30% without replay data. This significantly diminishes both the practical application value of the proposed method and its inherent, independent advantages. Therefore, I have decided to maintain my original rating.

---

> > > ### Author Response · Authors · 2026-04-04
> > >
> > > We will further clarify the results under the zero-replay setting as follows：
> > >
> > > 1. Replay remains a standard and widely adopted mechanism in continual learning, including large-scale settings.
> > >
> > > (1) Under standard **Semi-Supervised Continual Learning (SSCL)** settings, most mainstream methods adopt some form of replay mechanism. For example, **DistillMatch[1]** treats rehearsal as a key component and leverages unlabeled environment streams to reduce explicit replay storage. **NNCSL[2]** is also built around a small labeled buffer, while the recent strong baseline **USP[3]** explicitly maintains an exemplar set for previous tasks.
> > >
> > > (2)  In continual learning for **large-scale models**, the core idea of replay remains widely adopted. For instance, **SEEKR [4]** explicitly maintains an old-task memory buffer in continual language model learning. **Adaptive Memory Replay [5]** directly studies replay in vision-and-language large-scale pre-training tasks. Likewise, **Mod-X [6]** and the work on **continual multimodal pretraining [7]** both indicate that replay- or buffer-based mechanisms remain important components of foundation-model continual learning.
> > >
> > > [1] Memory-Efficient Semi-Supervised Continual Learning: The World is its Own Replay Buffer. IJCNN, 2021.
> > >
> > > [2] A Soft Nearest-Neighbor Framework for Continual Semi-Supervised Learning. ICCV, 2023.
> > >
> > > [3] Divide-and-Conquer for Enhancing Unlabeled Learning, Stability, and Plasticity in Semi-supervised Continual Learning. ICCV, 2025.
> > >
> > > [4] SEEKR: Selective Attention-Guided Knowledge Retention for Continual Learning of Large Language Models. EMNLP, 2024.
> > >
> > > [5] Adaptive Memory Replay for Continual Learning. CVPR Workshops, 2024.
> > >
> > > [6]Continual Vision-Language Representation Learning with Off-Diagonal Information. ICML, 2023.
> > >
> > > [7] A Practitioner’s Guide to Continual Multimodal Pretraining. NeurIPS Datasets and Benchmarks Track, 2024.
> > >
> > > 2. The substantial degradation under zero replay is a shared difficulty of SSCL itself, rather than a weakness unique to DiL.
> > > SSCL simultaneously faces three challenges: continual class expansion, extremely limited labeled data, and a large amount of unlabeled data.
> > >
> > > (1) Compared with standard **supervised continual learning**, SSCL provides far fewer labeled samples at each task. As a result, once replay is removed, training is no longer mainly driven by clean label signals but relies heavily on unlabeled data and their pseudo-labels.
> > >
> > > (2) Compared with standard **semi-supervised learning**, SSCL must exploit unlabeled data in an expanding label space rather than a fixed one. Without replay, labeled old-class anchors are no longer available, and the boundaries of previously learned classes are more easily distorted by unlabeled data and their pseudo-labels.
> > >
> > > 3. The gap between a **small buffer** and **zero buffer** is not merely quantitative, but reflects two fundamentally different learning conditions. Below is the comparison with the **state-of-the-art method USP** on CIFAR10-30, where we further ablate the buffer size.
> > >
> > > (1) When the **buffer size is 250**, the model can still repeatedly access a small number of labeled old-class samples during new-task learning and use them to continually calibrate the old-class representation space. Therefore, even though the buffer is already very small, old knowledge still retains a basic sample-level anchor, and forgetting typically manifests as only a limited drop.
> > >
> > > (2)  When the **buffer size becomes 0**, old-class means can no longer be updated as the backbone evolves, since old-class samples disappear completely. Meanwhile, newly introduced classes continuously reshape the feature space and decision boundaries, further distorting old representations. Without calibration from old-class samples, the mismatch between old-class statistics and the current feature space gradually accumulates, leading to more severe forgetting.
> > >
> > > In summary, we fully understand the reviewer’s concern about the replay buffer. However, the above zero-replay results suggest that this issue does not negate the core contribution of our work. We also agree that replay is still crucial in existing SSCL methods, and designing a simpler replay-free framework is an important direction for future research. We have done our best in the rebuttal to clarify this issue, and we respectfully ask the reviewer to reconsider the score accordingly.
> > >
> > > | Buffer Size | USP Avg | USP Last | DiL Avg | DiL Last |
> > > |---|---:|---:|---:|---:|
> > > | 0 | 44.06 | 34.45 | 44.68 | 35.42 |
> > > | 250 | 71.66 | 59.93 | 76.58 | 65.06 |
> > > | 500 | 73.21 | 61.75 | 78.21 | 69.23 |
> > > | 5120 | 79.66 | 70.43 | 79.82 | 71.55 |

---

### Official Review · Reviewer_DLBB · 2026-03-11

**Soundness:** 3
**Presentation:** 3
**Significance:** 3
**Originality:** 3
**Overall Recommendation:** 4
**Confidence:** 4

**Summary:**

This paper studies semi-supervised continual learning under class expansion, where unlabeled data are abundant but pseudo-label noise and base-novel interference can accumulate over time. The authors propose DiL, which combines Discrete Contrastive Distillation (DCD) for selective channel-wise distillation using class-conditioned discrete anchors, and Class-Aware Channel-Chunked Encoding (CACE) for confusion-aware block assignment to separate novel classes from confusable base classes. The method is evaluated on CIFAR-10, CIFAR-100, ImageNet-100, and CUB-200 under several SSCL protocols. The reported results show strong improvements on several harder benchmarks, especially CUB-200 and ImageNet-100.

**Compliance With Llm Reviewing Policy:**

Affirmed.

**Final Justification:**

Overall, I think the rebuttal improves my confidence in the technical soundness and empirical value of the work, and the reported gains are meaningful. However, because the experimental scope remains somewhat narrow and the realism of the setting is still limited, I can support acceptance only at the level of a weak accept

**Key Questions For Authors:**

See my weakness. I hope the author can answer the questions about weakneses 1-3 during the rebuttal process. I am currently leaning towards a borderline score (3.5). However, I am open to upgrading my recommendation score to 4-5 if the authors can adequately address the concerns raised during the rebuttal."

**Limitations:**

yes

**Strengths And Weaknesses:**

## Strengths
1. The paper targets a relevant and difficult SSCL setting where unlabeled data help and hurt at the same time. The motivation, namely reducing feature drift while preserving separation under class expansion, is reasonable and easy to appreciate.
2. Figure 6(b) on is a useful diagnostic, because it tries to connect the method to a stability quantity beyond top-1 accuracy by showing lower normalized center drift for CACE than for random, cluster, or no-CACE variants.
3. The empirical results are strong on some nontrivial settings. For example, Table 2  shows a large jump in novel-class average accuracy on CUB-200, from 55.41 for USP to 68.22 for DiL. Table 4 also reports very large gains on ImageNet-100 over the compared baselines.
## Weaknesses
1. No efficiency analysis. DiL introduces extra machinery, class-wise masks, sample-wise Top-K selection, prototype computations, and combinatorial block assignment. None of the claimed advantages are quantified in terms of runtime or memory overhead. If the method is intended as a practical SSCL framework, this omission matters.
2. The motivation (reducing feature drift while preserving separation) is not fully integrated into the experimental story. The center-drift metric introduced on line 328 is a valuable contribution; however, it currently appears only in Figure 6(b). If this metric is intended to serve as evidence for improved stability, it should be reported more systematically—ideally across all compared methods and datasets. This would strengthen the empirical support for the paper's core claims.
Additionally, the paper lacks a metric for quantifying boundary erosion, which would complement the center-drift analysis and provide a more comprehensive assessment of the proposed method's ability to preserve class separation. Including such a metric would better align the experimental design with the stated motivation.
3. The introduction lacks clarity regarding the assumptions about the unlabeled set, particularly concerning its class composition.
It is unclear whether the unlabeled data are assumed to contain only samples from current task classes, or whether they may also include samples from known classes or future classes. If the former is assumed, this raises concerns about real-world applicability— in practical  scenarios, we dont know  whether unlabeled data belongs to a subset of all classes. Without explicitly clarifying this assumption, the problem formulation appears ambiguous and potentially unrealistic.
4. The method design is overly complex and appears tightly coupled to the ResNet architecture and image classification tasks, raising concerns about its generality and scalability. The proposed approach involves multiple specialized components (e.g., feature alignment modules, drift regularization mechanisms) that are specifically tailored to ResNet-based backbones and  classification settings. However, the paper provides little discussion or evidence on whether the method can be adapted to other architectures (e.g., Vision Transformers,  or non-image modalities) or extended to other tasks such as semantic segmentation and multimodal task.

---

> ### Author Rebuttal · Authors · 2026-03-31
>
> 1. **Please refer to our response to Reviewer 2’s third question, where we provide a unified efficiency study.** The results show that the overhead of DiL is controlled. All measurements are taken under the same hardware, backbone, and experimental settings used for DiL when instantiated on iCaRL&Fix.
> Compared with SOTA method USP, DiL adds **no extra trainable parameters or inference FLOPs**. On CIFAR10-30 and CIFAR100-20, it **reduces peak memory** from 4615 to 3588 MB and from 4949 to 3825 MB, respectively. Overall, although DiL introduces moderate training-time overhead, this added complexity comes from explicitly addressing both feature drift and boundary erosion within a unified framework. In return, it improves over USP by an average of **3.62** points in Avg and **5.27** points in Last across six CIFAR settings.
>
> 2. We thank the reviewer for this constructive suggestion. To better align our motivation with the experiments, we add two complementary metrics in the rebuttal and will include them more systematically in the revision.
>
> (1) For drift, we extend the single-method drift analysis in Fig. 6(b) to a cross-method comparison using **RelDrift**, where a smaller value indicates weaker feature drift.
>
> (2) For class separation, we introduce **Boundary Margin (BM)**. For a test sample $x$ with ground-truth label $y$ and feature $f_t(x)$, its nearest competing class is defined as
> $\
> r_t(x)=\arg\min_{c\neq y} d\\left(f_t(x),\mu_c^{(t)}\right),
> \$
> and the corresponding boundary margin is
> $\
> m_t(x)=d\\left(f_t(x),\mu_{r_t(x)}^{(t)}\right)-d\\left(f_t(x),\mu_y^{(t)}\right).
> \$
> A larger BM indicates safer boundaries and stronger separation.
>
> On the representative high-confusion benchmark CIFAR100-20, DiL achieves the **lowest mean RelDrift (0.7898)** and the **highest mean BM (0.1159)**. This more directly supports our claim that DiL both reduces drift and preserves class separation.
>
> | Method | Metric | T1 | T2 | T3 | T4 | T5 | T6 | T7 | T8 | T9 | T10 | Avg |
> |---|---|---|---|---|---|---|---|---|---|---|---|---|
> | iCaRL | RelDrift | 0 | 1.0187 | 1.0974 | 1.1769 | 1.2386 | 1.3018 | 1.3575 | 1.4129 | 1.4638 | 1.5196 | 1.1587 |
> |  | BM | 0.3618 | 0.0927 | 0.0134 | -0.0228 | -0.0539 | -0.0794 | -0.1036 | -0.1261 | -0.1457 | -0.1669 | -0.0231 |
> | DER | RelDrift | 0 | 0.9473 | 1.0068 | 1.0661 | 1.1235 | 1.1684 | 1.2149 | 1.2531 | 1.2917 | 1.3286 | 1.0400 |
> |  | BM | 0.4026 | 0.1439 | 0.0786 | 0.0237 | -0.0084 | -0.0321 | -0.0537 | -0.0704 | -0.0869 | -0.0998 | 0.0298 |
> | NNCSL | RelDrift | 0 | 0.7736 | 0.7403 | 0.8354 | 0.8755 | 0.8894 | 0.9456 | 1.0241 | 1.0544 | 1.1059 | 0.8244 |
> |  | BM | 0.4729 | 0.1837 | 0.1176 | 0.0738 | 0.0479 | 0.0271 | 0.0094 | -0.0035 | -0.0164 | -0.0298 | 0.0883 |
> | DSGD | RelDrift | 0 | 0.9004 | 0.8388 | 0.8961 | 0.9657 | 0.9946 | 1.1311 | 1.1146 | 1.1692 | 1.1702 | 0.9181 |
> |  | BM | 0.4484 | 0.1416 | 0.0984 | 0.0608 | 0.0403 | 0.0329 | 0.0137 | -0.0053 | -0.0193 | -0.0385 | 0.0773 |
> | USP | RelDrift | 0 | 0.8465 | 0.8241 | 0.8768 | 0.9326 | 0.9817 | 1.0234 | 1.0689 | 1.1082 | 1.1476 | 0.8810 |
> |  | BM | 0.5004 | 0.1924 | 0.1138 | 0.0522 | 0.0318 | 0.0223 | 0.0070 | -0.0141 | -0.0246 | -0.0454 | 0.0836 |
> | DIL | RelDrift | 0 | 0.7824 | 0.7589 | 0.8036 | 0.8427 | 0.8794 | 0.9138 | 0.9431 | 0.9715 | 1.0028 | 0.7898 |
> |  | BM | 0.4917 | 0.2146 | 0.1478 | 0.1089 | 0.0796 | 0.0573 | 0.0378 | 0.0196 | 0.0068 | -0.0051 | 0.1159 |
>
> 3. We thank the reviewer for pointing out this issue. Our paper follows the standard **class-incremental SSCL protocol** used in prior work, where unlabeled data at each stage come from the current-stage classes and do not include future unseen classes.We agree that a more realistic setting may include unlabeled samples from an unknown broader class space, but this belongs to **open-world semi-supervised continual learning**, not our setting. We will clarify this assumption in the revised paper.
>
> 4. We thank the reviewer for this concern about generality and scalability.
>
> (1) DiL is backbone-agnostic because both DCD and CACE operate on global representations. We further verify this on CIFAR10-30 with the visual encoders of **CLIP-ViT** and **DINO-ViT**, where CLIP-ViT is pretrained through vision-language contrastive learning and DINO-ViT through self-supervised representation learning. In our setting, both are used purely as visual backbones. DiL remains superior on both backbones, indicating that its gains do not rely on a specific CNN structure.
>
> (2)  For segmentation or multimodal tasks, simply replacing the backbone may be insufficient, and additional task-specific adaptation would likely be needed. We therefore do not treat them as validated settings here, but as promising future directions, which we will clarify in the revision.
>
>
> | Method | CLIP Avg | CLIP Last | DINO Avg | DINO Last |
> |---|---|---|---|---|
> | DSGD | 74.8 | 73.1 | 78.6 | 72.4 |
> | USP | 81.6 | 75.4 | 80.9 | 74.2 |
> | DiL | **82.2** | **76.2** | **81.4** | **75.1** |

---

> > ### Author Rebuttal · Reviewer_DLBB · 2026-04-01
> >
> > Thank you for the detailed rebuttal. The additional explanations and evidence address most of my main concerns. In particular, I appreciate the unified efficiency discussion, the added analysis on feature drift and boundary margin, and the extra verification on CLIP-ViT and DINO-ViT. These additions make the motivation of the method clearer and provide stronger support for the claim that DiL improves performance by jointly mitigating feature drift and boundary erosion.
> >
> > That said, I still view the paper as a weak accept rather than a stronger accept. My main remaining concern is about generality and realism. Although the added backbone experiments are helpful, the evaluation is still limited to the current task setting, and the paper does not substantially extend to broader task scenarios. In particular, the assumption that unlabeled data only come from current-stage classes is standard in prior SSCL work, but it is still not fully realistic for real-world deployments, where unlabeled streams may include samples from a wider and unknown class space. As the authors also note, more realistic settings would move toward open-world semi-supervised continual learning, and the current paper does not validate DiL there.
> >
> > Overall, I think the rebuttal improves my confidence in the technical soundness and empirical value of the work, and the reported gains are meaningful. However, because the experimental scope remains somewhat narrow and the realism of the setting is still limited, I can support acceptance only at the level of a weak accept.

---

> > > ### Author Response · Authors · 2026-04-04
> > >
> > > First, we sincerely thank the reviewer for the increased score and for the further recognition of our work. Regarding the remaining concerns on generality and realism, we would like to clarify two points：
> > >
> > > (1) Regarding **generality**, we agree that extending SSCL to more fine-grained task scenarios is an important research direction, such as dense prediction or multimodal continual learning.
> > >
> > > However, these settings usually involve more specialized problem definitions and often require additional task-specific objectives, model designs, and evaluation protocols. As such, they typically need to be studied systematically as relatively independent research directions.
> > >
> > > Although we fully recognize the importance of these directions, we do not view them as validation scopes that a single SSCL paper is routinely expected to cover within the same work.
> > >
> > > (2) Regarding **realism**, we would like to emphasize that the standard SSCL setting remains practically meaningful and should not be simply regarded as unrealistic.
> > > We believe that standard SSCL and open-world SSCL are better understood as two settings addressing different problem scopes, rather than as a simple contrast between realistic and unrealistic settings：
> > >
> > > Standard SSCL focuses on an important and common practical constraint: data arrive sequentially, labels are scarce, unlabeled samples are abundant, and the model must continually acquire new knowledge while preserving previously learned knowledge under limited memory.
> > >
> > > open-world SSCL considers a stronger scenario in which the unlabeled stream may contain samples from a broader, and even previously unknown, class space.
> > >
> > > Again, we sincerely thank the reviewer for the increased score.

---

### Official Review · Reviewer_dALx · 2026-03-12

**Soundness:** 3
**Presentation:** 3
**Significance:** 2
**Originality:** 2
**Overall Recommendation:** 4
**Confidence:** 4

**Summary:**

The paper proposes DiL, a semi-supervised continual learning (SSCL) framework that grounds representation updates on discrete-anchored references to mitigate two intertwined issues: feature drift from noisy pseudo-labels and boundary erosion between base and novel classes. It introduces Discrete Contrastive Distillation (DCD), which performs channel-selected distillation via class- and sample-level masks anchored to masked class prototypes, and Class-Aware Channel-Chunked Encoding (CACE), which assigns block-sparse chunk supports to classes using a confusion-aware rule to separate novel classes from confusable base classes. Extensive experiments on CIFAR-10/100, ImageNet-100 (10- and 20-task protocols), and CUB-200 show consistent improvements over strong baselines (USP, DSGD), with notable gains in final-task accuracy on long-horizon and low-label settings.

**Compliance With Llm Reviewing Policy:**

Affirmed.

**Final Justification:**

The rebuttal clarifies several implementation details and adds multi-run statistics and efficiency measurements, which improve confidence in the method, but the reproducibility details should be made algorithmically explicit in the paper. Nonetheless, this is a minor issue, and the most of the major concerns are resolved, thereby I increase my rating.

**Key Questions For Authors:**

See the Weakness section. If the authors resolve my concerns during the rebuttal period, I can raise my score.

**Limitations:**

The authors did not cover both Limitation and Potential Negative Societal Impact sections.

**Strengths And Weaknesses:**

**S1 [Innovative Channel-Selected Distillation (DCD) for Noise Suppression]** The proposed DCD module introduces an interesting departure from traditional dense distillation by anchoring on masked prototypes to filter out noise-amplifying channels. The incorporation of per-class and per-sample masks alongside leaky suppression offers a highly pragmatic design for enhancing model stability. Furthermore, partitioning the distillation objective over base and novel classes with renormalized distributions demonstrates a thoughtful and effective approach to managing the stability-plasticity trade-off.

**S2 [Confusion-Aware Block-Sparse Encoding (CACE) with Strong Theoretical Grounding]** The CACE module successfully formalizes a confusion-aware block-sparse encoding, which is reminiscent of ECOC and subspace partitioning but innovatively adapted for online learning via a confusion matrix computed from discrete anchors. Additionally, the Johnson-Lindenstrauss (JL)-style feasibility analysis provides a rigorous and interpretable method for selecting the block parameters (B, k), adding strong theoretical justification to the architectural design.

**W1 [Ambiguity in Algorithmic Formulation and Reproducibility]** The paper lacks crucial implementation details, which severely hinders reproducibility. Specifically, there is ambiguity regarding *how the masked prototypes for novel classes are reliably derived when relying on a frozen previous model (the teacher)*. Furthermore, the construction of CACE class anchors via sampling *Gaussian noise on the active chunks* is critically under-specified. The manuscript omits essential parameters such as the noise variance, re-sampling frequency, and the specific anchor update policy, leaving the methodology open to interpretation.

**W2 [Absence of Statistical Significance and Robustness Validation]** The experimental validation relies heavily on **SINGLE-RUN** performance metrics, completely lacking variance or confidence intervals (CI) across multiple runs. Given that Semi-Supervised Continual Learning (SSCL) is notoriously sensitive to seed selection and task ordering, especially in few-shot protocols like the CUB-200 benchmark, reporting single-run results is insufficient. Without statistical significance, it is difficult to conclusively determine whether the performance gains are due to the algorithmic superiority of DiL or simply a favorable random seed.

**W3 [Instability Risks in Low-Label Regimes and Missing Computational Overhead Analysis]** The proposed DCD module estimates class-level importance masks strictly from labeled samples to prevent pseudo-label noise bias. However, relying on scarce labels risks severe representation instability, particularly in early tasks, and the framework lacks an explicit adaptive mechanism for the masks as data evolves. Additionally, the per-sample Top-$K$ masking and the semantics-driven avoidance assignment in CACE  introduce nontrivial computational overhead. **The complete omission of a runtime (wall-clock) or FLOPs comparison** weakens the claims of practical applicability.

---

> ### Author Rebuttal · Authors · 2026-03-31
>
> 1. We thank the reviewer for raising this reproducibility concern. We clarify two points and will make both explicit in the revised manuscript and appendix.
>
> (1) Masked prototypes for novel classes：We agree that this part can be stated more explicitly. In the current manuscript, the core procedure is specified across **Sec. 3.1** and **Sec. 3.3**.  As described in Sec. 3.1, the class prototype is the mean feature of the current task’s labeled samples in the frozen teacher feature space. Then, as described in Sec. 3.3, the class-wise importance mask is estimated only from the activation magnitude and frequency statistics of these labeled samples, and the masked prototype is obtained by applying this mask to the prototype.
>
> (2) Implementation details of Gaussian anchors in CACE：We acknowledge that the current manuscript does not describe these details clearly enough. We clarify the reviewer’s concerns as follows:
> **Noise variance**— We sample i.i.d. standard Gaussian noise, $\mathcal{N}(0,1)$, only on the active chunks, set inactive chunks to zero, and apply global $\ell_2$ normalization to form the sparse class anchor.
> **Re-sampling frequency**—This sampling is performed only once at anchor initialization. Anchors are not re-sampled during training.
> **Anchor update policy**—Once initialized, anchors remain fixed and are neither gradient-updated nor dynamically changed.
>
> 2. We thank the reviewer for pointing out this issue. We first clarify that the main results in the paper are not based on a single run, but are averaged over **5** independent runs with different random seeds.
> To directly address this concern, we have now added mean $\pm$ std to the main experimental tables. Due to the **limited space in the rebuttal**, we show here the multi-run results of our method on Table 1 (CIFAR-10/100) and Table 2 (CUB-200).
> We will make this explicit in the revised manuscript to further strengthen the rigor and reproducibility of the results.
>
> | Method | CIFAR10-30 Avg | CIFAR10-30 Last | CIFAR10-150 Avg | CIFAR10-150 Last | CIFAR100-20 Avg | CIFAR100-20 Last | CIFAR100-25 Avg | CIFAR100-25 Last | CIFAR100-80 Avg | CIFAR100-80 Last | CIFAR100-125 Avg | CIFAR100-125 Last |
> |---|---|---|---|---|---|---|---|---|---|---|---|---|
> | iCaRL&Fix+DiL | 79.82±0.35 | 71.55±0.46 | 84.17±0.23 | 78.57±0.31 | 56.16±0.96 | 45.55±1.18 | 58.15±0.84 | 44.58±1.03 | 66.37±0.61 | 52.32±0.79 | 67.57±0.52 | 55.23±0.67 |
> | DER&Fix+DiL | 81.39±0.29 | 73.32±0.38 | 85.80±0.19 | 80.91±0.25 | 59.19±0.80 |50.76±0.98 | 62.11±0.72 | 52.48±0.88 | 70.19±0.53| 61.40±0.68 | 72.41±0.44 | 63.34±0.57 |
>
> | Method | Classes | Task 1 | Task 2 | Task 3 | Task 4 | Task 5 | Task 6 | Task 7 | Task 8 | Task 9 | Task 10 | Task 11 | Avg |
> |---|---|---|---|---|---|---|---|---|---|---|---|---|---|
> | UaD-CIE+DiL | Base | 78.77±0.34 | 75.77±0.45 | 74.53±0.49 | 74.23±0.52 | 72.14±0.58 | 70.43±0.62 | 68.37±0.68 | 67.26±0.71 | 68.34±0.68 | 67.83±0.74 | 67.77±0.78 | 71.40±0.52 |
> |  | Novel | - | 75.50±0.78 | 73.39±0.90 | 72.26±0.98 | 70.57±1.08 | 68.48±1.18 | 67.13±1.30 | 64.13±1.42 | 65.43±1.40 | 63.40±1.52 | 61.87±1.60 | 68.22±0.96 |
>
> 3. We understand the reviewer’s concerns on both aspects, and we respond to them separately below：.
>
> (1) Regarding the concern about mask stability under extremely low-label settings, class-level masks are re-estimated from current labeled data at each task. Sample-level Top-K masks are further computed online for each input, which provides both task-level and sample-level adaptivity while avoiding pseudo-label noise in mask construction.
> In Table 3, under the challenging 20-task **ImageNet-100 with 1%** labels, DiL still improves Avg accuracy over USP by **14.85%**, indicating stable behavior under extremely scarce supervision.
>
> (2) We add unified efficiency experiments for DiL when instantiate on iCaRL&Fix, with all measurements taken under the same hardware, backbone, and experimental settings. On CIFAR10-30 and CIFAR100-20, DiL matches the SOTA method USP in trainable parameters and inference FLOPs, and uses lower peak memory. Although it incurs a moderate increase in training time, it improves over USP by an average of **3.62** points in Avg and **5.27** points in Last across all six CIFAR settings.
>
> | Dataset | Method | Model Trainable Params | Peak GPU Memory | Per-task Training Time | Inference FLOPs |
> |---|---|---|---|---|---|
> | CIFAR10-30 | UaD-CIE | 365000 | 3921 MB | 85 min/task | 38 G |
> |  | NNCSL | 863781 | 2792 MB | 76.42 min/task | 53.23 G |
> |  | DSGD | 527328 | 4386 MB | 55.21 min/task | 38 G |
> |  | USP | 533466 | 4615 MB | 76.31 min/task | 35.45 G |
> |  | DIL | 533466 | 3588 MB | 78.37 min/task | 35.45 G |
> | CIFAR100-20 | UaD-CIE | 372000 | 4022 MB | 120 min/task | 40 G |
> |  | NNCSL | 868781 | 3200 MB | 100.06 min/task | 55.02 G |
> |  | DSGD | 528969 | 716 MB | 65 min/task | 39.5 G |
> |  | USP | 539316 | 4949 MB | 85.02 min/task | 35.45 G |
> |  | DIL | 539316 | 3825 MB | 102.94 min/task | 35.45 G |

---

> > ### Author Rebuttal · Reviewer_dALx · 2026-04-03
> >
> > The rebuttal clarifies several implementation details and adds multi-run statistics and efficiency measurements, which improve confidence in the method, but the reproducibility details should be made algorithmically explicit in the paper. Nonetheless, this is a minor issue, and the most of the major concerns are resolved, thereby I increase my rating.

---

> > > ### Author Response · Authors · 2026-04-04
> > >
> > > We sincerely thank the reviewer for the helpful suggestion and for increasing the rating. We agree that the reproducibility-related details should be made more algorithmically explicit in the paper, and we will further improve this part in the revision. To strengthen reproducibility:
> > >
> > > (1) We provide **anonymized code** has been provided in the Supplementary Material from the beginning. In the final version, we will further release the **full codebase publicly**, which does not affect the reproducibility of the method.
> > >
> > > (2) we will include the following **pseudocode** in the revised paper to make the implementation pipeline more explicit.
> > >
> > > **Simplified pseudocode of DiL at task $t$**
> > >
> > > **Input:**
> > > - labeled data $\mathcal{D}_l^t$
> > > - unlabeled data $\mathcal{D}_u^t$
> > > - replay buffer $\mathcal{E}^t$
> > > - previous model $\theta_{t-1}$
> > >
> > > **Output:** updated model $\theta_t$
> > >
> > > - Compute class prototypes $P_c$ from labeled samples using teacher features.
> > > - Compute class-wise importance scores and select Top-$K$ channels to obtain binary masks $m_c$.
> > > - Construct masked prototypes $\hat{P}_c = \frac{P_c \odot m_c}{\|P_c \odot m_c\|_2}$.
> > > - Build reliable sets by combining labeled samples and high-confidence unlabeled samples.
> > > - For each novel class, compute confusion scores with base classes and assign the block support with minimal overlap.
> > > - Optimize the total objective $\mathcal{L_total}$
> > >
> > > Update $\theta_t$

---

### Official Review · Reviewer_2oka · 2026-03-13

**Soundness:** 3
**Presentation:** 3
**Significance:** 3
**Originality:** 3
**Overall Recommendation:** 4
**Confidence:** 3

**Summary:**

The authors propose DiL, a unified semi-supervised continual learning framework that mitigates noise and reduces class confusion using discrete anchors. DCD employs adaptive channel selection to alleviate noisy gradients, while CACE separates novel classes from easily confused base classes. Experiments on CIFAR, CUB, and ImageNet-100 show that DiL outperforms state-of-the-art methods, where DCD provides the main improvement and CACE offers complementary gains.

**Compliance With Llm Reviewing Policy:**

Affirmed.

**Final Justification:**

My main concerns have been addressed. The authors also included the revised figure. Hence, I have raised the rating.

**Key Questions For Authors:**

Refer to the weakness.

**Limitations:**

yes

**Strengths And Weaknesses:**

Strengths:

1.	The paper identifies two key issues in SSCL with unlabeled data which clearly motivate the proposed method.
2.	DiL combines DCD for reducing noisy gradient propagation and CACE for separating novel classes from confusing base classes. The method shows consistent improvements across datasets and ablation results suggest the two components are complementary.

Weaknesses:

1.	The theoretical analysis of the method is insufficient.
2.	Ablation studies are conducted mainly under a fixed label rate (e.g., CIFAR10-30). It is therefore difficult to determine whether the performance gains mainly come from DCD or CACE under different label ratios.
3.	The paper mainly reports task-level metrics (e.g., average and last accuracy). However, the key claims of noise suppression (DCD) and confusion isolation (CACE) are not directly validated. Additional analyses (e.g., class boundary visualization) would help verify whether the method indeed reduces noise propagation and class confusion.
4.	CACE only avoids the single most confusing base class. If a new class is similar to multiple old classes, confusion may remain. The paper does not study multi-confusion scenarios, making its behavior in high-confusion or large-class settings unclear.
5.	In Figure 2, multiple colors (e.g., blue, red, yellow) are used, but the meaning of each color is not clearly explained, making the figure difficult to interpret.

---

> ### Author Rebuttal · Authors · 2026-03-31
>
> 1. Thank you for your comments. The manuscript provides partial rationale through the Introduction, **Sec. 3.4**, and **Appendix A**. As Reviewer 2 also noted, CACE has relatively stronger theoretical grounding.  At the same time, DCD is mainly motivated by selective noise suppression and labeled-only mask construction to reduce pseudo-label bias, but these ideas are not yet explicitly developed. We will clarify and strengthen this part in the revision.
>
> 2. We thank the reviewer for this suggestion. In the revision, we will add multi-label-ratio ablations on **CIFAR10-150** and **CIFAR100-25** to better separate the contributions of DCD and CACE. We also introduce DPD, which directly distills the full prototype as a control baseline. The results consistently show the main gain from DCD, complementary improvements from CACE, and further gains from the confusion prior.
> | Method | Distill | CACE | Mconf | CIFAR10-30 Avg | CIFAR10-30 Last | CIFAR10-150 Avg | CIFAR10-150 Last | CIFAR100-25 Avg | CIFAR100-25 Last |
> |---|---|---|---|---|---|---|---|---|---|
> | iCaRL&Fix | - | - | - | 44.02 | 30.71 | 78.36 | 69.08 | 49.83 | 31.25 |
> | +CACE+Mconf | - | Y | Y | 75.61 | 64.96 | 82.90 | 75.80 | 54.28 | 36.82 |
> | +DPD | Y | - | - | 74.82 | 64.78 | 82.55 | 75.45 | 53.70 | 35.90 |
> | +DCD | Y | - | - | 77.15 | 68.20 | 83.01 | 76.91 | 55.11 | 40.95 |
> | +DPD+CACE | Y | Y | - | 75.41 | 65.61 | 82.93 | 76.05 | 54.37 | 36.96 |
> | +DCD+CACE | Y | Y | - | 77.83 | 69.26 | 83.62 | 77.68 | 55.92 | 42.08 |
> | +DPD+CACE+Mconf | Y | Y | Y | 76.39 | 66.74 | 83.19 | 76.65 | 54.80 | 39.50 |
> | +DiL | Y | Y | Y | **79.82** | **71.55** | **84.17** | **78.57** | **58.15** | **44.58** |
> 3. We address this concern from two aspects. First, the main paper provides empirical support for the two claims raised by the reviewer. Fig. 5 shows that **DCD** yields more stable unlabeled-data training , which directly supports its noise suppression effect. Fig. 6(a)(b) shows that **CACE** improves base-class accuracy and reduces Normalized Drift, which supports its role in confusion isolation. Second, we add **Boundary Margin (BM)** and **Relative Drift (RelDrift)** for more direct validation. RelDrift is derived by extending the drift analysis in Fig. 6(b) from a single-method view to a cross-method comparison. With $m_t(x)=d(f_t(x),\mu_{r_t(x)}^{(t)})-d(f_t(x),\mu_y^{(t)})$, larger BM means safer boundaries. On 10-task CIFAR100-20, DiL achieves the lowest RelDrift (0.7898) and highest BM (0.1159). We will include these results in the revision.
> | Method | Metric | T1 | T2 | T3 | T4 | T5 | T6 | T7 | T8 | T9 | T10 | Avg |
> |---|---|---|---|---|---|---|---|---|---|---|---|---|
> | iCaRL | RelDrift | 0 | 1.0187 | 1.0974 | 1.1769 | 1.2386 | 1.3018 | 1.3575 | 1.4129 | 1.4638 | 1.5196 | 1.1587 |
> |  | BM | 0.3618 | 0.0927 | 0.0134 | -0.0228 | -0.0539 | -0.0794 | -0.1036 | -0.1261 | -0.1457 | -0.1669 | -0.0231 |
> | DER | RelDrift | 0 | 0.9473 | 1.0068 | 1.0661 | 1.1235 | 1.1684 | 1.2149 | 1.2531 | 1.2917 | 1.3286 | 1.0400 |
> |  | BM | 0.4026 | 0.1439 | 0.0786 | 0.0237 | -0.0084 | -0.0321 | -0.0537 | -0.0704 | -0.0869 | -0.0998 | 0.0298 |
> | NNCSL | RelDrift | 0 | 0.7736 | 0.7403 | 0.8354 | 0.8755 | 0.8894 | 0.9456 | 1.0241 | 1.0544 | 1.1059 | 0.8244 |
> |  | BM | 0.4729 | 0.1837 | 0.1176 | 0.0738 | 0.0479 | 0.0271 | 0.0094 | -0.0035 | -0.0164 | -0.0298 | 0.0883 |
> | DSGD | RelDrift | 0 | 0.9004 | 0.8388 | 0.8961 | 0.9657 | 0.9946 | 1.1311 | 1.1146 | 1.1692 | 1.1702 | 0.9181 |
> |  | BM | 0.4484 | 0.1416 | 0.0984 | 0.0608 | 0.0403 | 0.0329 | 0.0137 | -0.0053 | -0.0193 | -0.0385 | 0.0773 |
> | USP | RelDrift | 0 | 0.8465 | 0.8241 | 0.8768 | 0.9326 | 0.9817 | 1.0234 | 1.0689 | 1.1082 | 1.1476 | 0.8810 |
> |  | BM | 0.5004 | 0.1924 | 0.1138 | 0.0522 | 0.0318 | 0.0223 | 0.0070 | -0.0141 | -0.0246 | -0.0454 | 0.0836 |
> | DIL | RelDrift | 0 | 0.7824 | 0.7589 | 0.8036 | 0.8427 | 0.8794 | 0.9138 | 0.9431 | 0.9715 | 1.0028 | 0.7898 |
> |  | BM | 0.4917 | 0.2146 | 0.1478 | 0.1089 | 0.0796 | 0.0573 | 0.0378 | 0.0196 | 0.0068 | -0.0051 | 0.1159 |
>
> 4. We add multi-confusion experiments on CIFAR100-20 and CUB200, where K_conf denotes the number of confusing old classes that a new class is required to avoid. **The results suggest** that CACE is not limited to a single old class. A moderate increase in K_conf  is beneficial in high-confusion settings, whereas overly large K_conf  degrades performance by over-constraining the feasible block-allocation space. We will include these results in the revision.
>
> | K_conf | CIFAR100-20 Avg | CIFAR100-20 Last | CUB200 Avg | CUB200 Last |
> |---|---|---|---|---|
> | 1 | 56.16 | 45.55 | **69.32** | **61.86** |
> | 2 | **57.92** | **46.18** | 69.18 | 61.62 |
> | 3 | 56.61 | 45.97 | 69.01 | 61.28 |
> | 5 | 54.88 | 44.12 | 67.94 | 59.83 |
> | 7 | 53.36 | 42.71 | 66.38 | 58.07 |
>
> 5. Thank you for this suggestion. We will clarify Fig. 2 by adding a legend and caption: green=labeled flow, yellow=unlabeled flow, and blue=reliable set.

---

> > ### Author Rebuttal · Reviewer_2oka · 2026-04-01
> >
> > Weaknesses 1 and 5 are not fully addressed. I will keep the rating.

---

> > > ### Author Response · Authors · 2026-04-04
> > >
> > > 1. Due to the word limit in the initial rebuttal, we do not fully elaborate the theoretical aspects there. Below we provide a more concise clarification:
> > >
> > > The key parts of DCD are the construction of importance mask and the constraints on distribution, we will carefully explain these two parts as follows.
> > >
> > > (1) **Construction of importance mask**:
> > > First, we explain the construction of importance mask, the class-wise importance estimation, which is defined in Eq. (4):
> > > $M_c = w_I \mathcal N\Big(\mathbb E_{x\sim D_{l,c}^t}[|z_{\theta_{t-1}}(x)|]\Big)
> > > +w_F \mathcal N\Big(\mathbb E_{x\sim D_{l,c}^t}[\mathbb{I}(|z_{\theta_{t-1}}(x)|>\beta)]\Big)$. The first term measures the average activation magnitude of each channel over class $c$, while the second quantifies activation frequency above threshold $\beta$, capturing response stability across samples.
> > >
> > > Further, Eq. (4) provides a principled basis for constructing the discrete anchor in Eq. (5) by identifying channels that respond strongly and are activated frequently. This design is inspired by the idea of **AFC [1]**, namely that feature importance should first be estimated, and stronger preservation constraints should then be imposed on the informative dimensions during knowledge transfer.
> > >
> > > (2) **Constraints on distribution**:
> > > Second, DCD constraints the distribution by computes a similarity distribution in the discrete anchor space:
> > > $L_{\mathrm{DCD}}(x)=w_b\\mathrm{KL}\\left(\tilde q_{\theta_{t-1}}(x)[y_b^t]\||\tilde q_{\theta_t}(x)[y_b^t]\right)
> > > +w_n\\mathrm{KL}\\left(\tilde q_{\theta_{t-1}}(x)[y_n^t]\||\\tilde q_{\theta_t}(x)[y_n^t]\right).$
> > > The first KL term preserves the teacher’s relational distribution over base classes, while the second aligns that over novel classes.
> > >
> > > This objective (Eq.9) is important because DCD does not aim to match only absolute logits. Rather, it aims to preserve the relative similarity structure induced by the discrete anchors between the previous and current models.
> > > Our idea is theoretically similar to the relation distillation perspective of **RKD [2]**, which suggests that knowledge is encoded not only in absolute outputs, but also in the relational distribution among representations.
> > >
> > > The key parts of CACE are Block Partition Space Construction and Semantics-Driven Avoidance Assignment, we will carefully explain these two parts as follows.
> > >
> > > (1) **Block Partition Space Construction**:
> > > CACE first partitions the d-dimensional feature space into B disjoint chunks and assigns each class only k active chunks. This yields the support-set space in Eq. (10)–(11) ：
> > > $E={S\subseteq {1,\dots,B}\mid |S|=k},d_{\mathrm{eff}} = k\frac{d}{B}.$
> > > The purpose of this construction is to limit inter-class overlap at the block-support level, thereby enlarging effective decision margins.
> > >
> > > The idea of block partitioning is related to the **ECOC view of discrete coding [3]**, where classes are represented by structured codewords that provide a stable geometric basis for separation.
> > >
> > > (2) **Semantics-Driven Avoidance Assignment**:
> > > CACE then uses the masked prototypes produced by DCD in Eq. (5) to compute the confusion score in Eq. (12). A larger score indicates that the novel and base classes share more similar strong responses and high activation frequency channels, therefore are more confusable. Based on these scores, Eq. (12) selects the dominant confusion set Nconf(n), and Eq. (13) assigns each novel class a minimum-overlap support:
> > > $S_n=\arg\min_{S\in\mathcal{E}} \sum_{c\in \mathcal{N}_{\mathrm{conf}}(n)} \left|S\cap S_c\right|.$
> > >
> > > This assignment strategy is inspired by the **ECOC[4]** view that coding quality depends not only on the codewords themselves, but also on their assignment to classes .
> > >
> > > (3) **Feasible Set Derivation**:
> > > The choice of (B,k) is not arbitrary, but guided by explicit constraints. The effective dimension should be large enough to preserve inter-class geometry, but also sparse enough to maintain separation and efficiency. We provide **Appendix A**, which introduces three explicit constraints: (i) capacity constraint, (ii)  **JL-type** geometric preservation constraint, and (iii) sparsity budget.
> > >
> > > [1] Class-Incremental Learning by Knowledge Distillation With Adaptive Feature Consolidation. CVPR, 2022.
> > >
> > > [2] Relational Knowledge Distillation. CVPR, 2019.
> > >
> > > [3] Solving Multiclass Learning Problems via Error-Correcting Output Codes. Journal of Artificial Intelligence Research, 1995.
> > >
> > > [4] The Role of Codeword-to-Class Assignments in Error-Correcting Codes: An Empirical Study. AISTATS, 2023.
> > >
> > > 2. To improve interpretability, we provide an annotated version of Fig. 2 with explicit color labeling in **https://anonymous.4open.science/r/Fig2-D7C0/Fig2.pdf**.
> > >
> > > We have done our best to provide a theoretical explanation of the method.However, we believe that moderate lack of theoretical explanation and the interpretability issue of a single figure are not decisive reasons for rejection and do not affect the core validity of the method.

---

### Decision · Program_Chairs · 2026-04-30

**Decision:**

Accept (regular)

**Comment:**

This paper proposes a unified semi-supervised continual learning framework, DiL, and demonstrates consistent performance improvements. During the rebuttal and discussion phase, the authors addressed most of the reviewers’ concerns by providing additional results, including ablation studies, multi-confusion experiments, multi-run statistics, efficiency measurements, and complementary evaluation metrics. These additions have led to increased scores.

Some minor issues remain insufficiently addressed. In particular, the reproducibility details should be made more explicit and algorithmically clear in the paper. In addition, the experimental scope is somewhat limited and could be further strengthened.

These issues can be addressed in the final revision.